psychology

infant categorization, computational models, memory, infant learning

**Author for correspondence:**
Nadja Althaus
e-mail: n.althaus@uea.ac.uk

# Infant categorization as a dynamic process linked to memory

Nadja Althaus[1,†], Valentina Gliozzi[2,†], Julien Mayor[3] and Kim Plunkett[4]

[1]School of Psychology, University of East Anglia, Norwich Research Park, Norwich NR4 7TJ, UK
[2]Center for Logic, Language, and Cognition, Department of Computer Science, University of Torino, C.so Svizzera 185, 10149 Torino, Italy
[3]Department of Psychology, University of Oslo, PO Box 1094, Blindern, 0317 Oslo, Norway
[4]Department of Experimental Psychology, University of Oxford, South Parks Road, Oxford OX1 3UD, UK

NA, 0000-0003-4888-1508

Recency effects are well documented in the adult and infant literature: recognition and recall memory are better for recently occurring events. We explore recency effects in infant categorization, which does not merely involve memory for individual items, but the formation of abstract category representations. We present a computational model of infant categorization that simulates category learning in 10-month-olds. The model predicts that recency effects outweigh previously reported order effects for the same stimuli. According to the model, infant behaviour at test should depend mainly on the identity of the most recent training item. We evaluate these predictions in a series of experiments with 10-month-old infants. Our results show that infant behaviour confirms the model's prediction. In particular, at test infants exhibited a preference for a category outlier over the category average only if the final training item had been close to the average, rather than distant from it. Our results are consistent with a view of categorization as a highly dynamic process where the end result of category learning is not the overall average of all stimuli encountered, but rather a fluid representation that moves depending on moment-to-moment novelty. We argue that this is a desirable property of a flexible cognitive system that adapts rapidly to different contexts.

## 1. Introduction

The ability to form categories is a core cognitive ability that allows us to structure the world. 'Categorization' implies treating distinct items as similar, requiring the same response [1]. When infants begin to understand the structure of the world, the vast majority

†These authors contributed equally to this study.

of stimuli are novel and categories need to be learned. Many studies have investigated how the ability to group objects changes throughout infancy and what factors influence this process, such as the feature distribution provided in the input [2–4], the presence of verbal labels [5–7] or the function of an object ([8], for a review, see [9]). Models of categorization agree that learning is information aggregation, but how exactly a mental representation is constructed from the encountered exemplars remains an open question. Infant category learning is often studied in familiarization paradigms, where a sequence of objects from the target category is presented as a learning phase, prior to one or several test trials during which the established representation is probed. Such familiarization procedures can be used to tap into already existing category representations [10,11], where the assumption is that they activate previously acquired knowledge. The focus in the current paper is, by contrast, on learning novel categories where prior representations cannot be used for grouping objects. The received wisdom under this scenario is that infants extract regularities of the familiarized stimulus set over time, constructing an abstract category representation or even more than one, if the familiarization set consists of several clusters. At test infants demonstrate the formation of such categories by exhibiting novelty preference for an out-of-category (OOC) item over a new item from a familiarized category [3,12–15]. Implicit in this logic is the idea that the category representation is, by that point, fixed, or at least converging to a category centroid.

Here, we propose by contrast that the category representation is much more fluid, affected to a great deal by recent memory, and that infant looking behaviour reflects these moment-to-moment shifts in category representation. We argue that this is the very property that allows for continuous learning and adaptation to varying context, allowing the human cognitive system to be highly flexible.

In real life, in contrast to laboratory experiments on category learning, there is no segregation of learning phase and test: every time an object is encountered, a category needs to be activated, and this process depends on similarity to previously encountered exemplars, as well as proximity in time. For example, if a child meets a horse in the meadow after an ordinary day during which she has encountered the neighbour's dog and some pigeons in the park, the horse may be interesting on the basis of its sheer size and general shape. By contrast, if the same horse is encountered on a day the child was taken to a horse farm, encountering the horse in the meadow will activate recent memories of the animals seen during the day—and they are likely to be very detailed. This might highlight, for example, the fact that this particular horse has a white blaze—in particular if the other horses did not have such a mark. In other words, how remarkable or interesting a particular exemplar is depends entirely on context.

We consider the events described above—encountering a horse after having seen other members of the same category—as a form of priming: if similar items have been encountered in the recent past, a corresponding mental representation is easily activated—but it is biased towards the particular recent experience. As we shall argue, this is essentially what occurs in a familiarization procedure. When the test trial is encountered, the representation that is activated is not a fixed entity with a centroid and a boundary. It is instead a representation shaped by the stimuli that have been seen during familiarization, and partially activated by the most recent one.

Here, we demonstrate a recency effect in a self-organizing map model of infant categorization, which allowed us to generate precise predictions for the behaviour of 10-month-old infants. Recency effects are well-documented in the adult [16,17] and infant [18] literature, but here we link these memory effects explicitly to category formation. As we shall see, the infant data correspond closely to the model's prediction: after familiarization with the same set of eight stimuli, a preference for a category outlier is either achieved or not, depending on the final familiarization stimulus that precedes the test trial. The model allows us to inspect how categories are encoded by the model. Observing the internal category representation in the model directly provides insight into the mechanisms at work.

In summary, we argue here that category activation does not merely involve invoking a pre-existing abstract representation, but is inherently linked to memory. Consequently, differences in novelty preference across infants may often not merely represent noise in the system, but reflect a tight coupling between infants' most recent experiences and a flexibly adapting category representation. The central role of memory in human category learning, specifically in adults, has been highlighted previously (e.g. [19,20]). However, our view specifically focuses on memory effects in infant familiarization as an online process.

Our view of categorization as closely intertwined with memory processes is consistent with other accounts of categorization that take seriously the emergence of categories in real time and emphasize the role of the encoding of individual stimuli versus the crystallization of a category representation from these occurrences [21–23]. Our model of categorization is convergent with approaches that follow a dynamic systems view of cognition [24–26], but adopts a more abstract

perspective, in which concepts respond to sets of features, and this response changes across discrete learning trials.

# 2. Background: infant categorization and memory

As described above, category learning in infants is often assessed using familiarization with a category involving several distinct exemplars, followed by a novelty preference test. Here, infants are presented with a novel within-category test stimulus and a novel OOC item simultaneously. A preference for the OOC item is taken as an index of successful categorization [27–32]. This builds onto earlier paradigms first established by Fantz [30], who found that infants presented with novel and familiar images side by side rapidly progress to preferring novel over repeated stimuli. An alternative procedure involves habituation to criterion, i.e. the continued presentation of stimuli until looking has decreased to a specific point, usually specified as a proportion of looking duration measured for the first item(s). In this kind of paradigm infants are often presented with single items at test, i.e. 'preference' is not established on a single display but by evaluating how looking time changes across trials. Here, a novel item from the familiar category is presented prior to presenting the OOC item, and the expectation is that looking will not increase for the novel item from the familiar category, but for the OOC item [33]. This is often used where examination of toy objects is employed as a metric instead of looking time (e.g. [22,34,35]). The logic of the novelty preference procedure rests on the interpretation of looking time as a measure of surprise. Longer looking at the OOC item suggests that it is less similar to the infant's internal category representation of the familiarization stimuli than the within-category item, i.e. less expected. In fact, Strauss [36] argued that infants formed categories on the basis of prototypes corresponding to the average of the stimuli they experienced.

Different approaches investigating infant categorization have considered memory as a core factor. One of these is Westermann & Mareschal's [37] model. These authors provide a dual-systems approach to categorization, where a fast-learning 'hippocampal' system is linked with a slow-learning 'cortical' system (both implemented as auto-encoders but trained with different learning rates). Their model successfully simulated results such as the development of global before basic-level categories that is observed in manual object exploration tasks and generalized imitation studies (e.g. [10]), which has given rise to the controversial theory that perceptual and conceptual categorization rely on two different processes. Effects of prior knowledge on familiarization results [38] are simulated in the same model. As such, this approach provides an elegant solution to the controversial question of how perceptual and conceptual categorization relate to each other that does not require two separate processes but uses a single system. While Westermann and Mareschal's approach places an emphasis on the role of memory in categorization, the focus in their model is explicitly on differences between rapid online processes and slower processes integrating new information with existing knowledge. In the present paper, by contrast, we direct our attention specifically to memory effects in online category learning. While item memory during familiarization has been investigated in the literature [39–42], many of these approaches did not address the nature of the category being learned but were instead focused on establishing whether or not infants showed evidence for remembering a particular exemplar. By contrast, Mareschal *et al.* [43] examined how infants' object examination time during a categorization task differed as a function of similarity between individual items. They provided evidence for an effect of the previous item on examination time for the current item. If the second item in the sequence was dissimilar to the first, then seven- to nine-month-olds' examination times were longer than if the second item was similar to the first. This effect appears to reflect online processes as discussed above, and is consistent with the hypothesis that memory for the most recent item plays a large role. A series of studies by Oakes and colleagues provided further insight into context effects during infants' learning. Oakes & Ribar [23] as well as Kovack-Lesh & Oakes [21] showed that presenting stimuli in pairs instead of individual items affects categorization behaviour. Four-month-old infants, for instance, formed more exclusive categories in the former than in the latter case. Since the formation of a more narrow category indicates that more detailed features of the viewed items are represented, this implies that studying items side by side benefits the identification or extraction of such relevant features. Once again, the way infants inspect a new item is shown to be context-dependent.

Gliozzi *et al.* [44] presented a neurocomputational model of familiarization and labelling (simulating infant results which demonstrated that labels modulate category boundaries [7]). This model suggested the existence of sequence effects [45]. On this basis, Mather & Plunkett [46] investigated item succession

in detail. They manipulated the 'Euclidean distance' between successive items across a set of eight familiarization items, meaning that infants in one group saw a sequence in which successive items were likely to be quite similar (termed 'low Euclidean distance' sequences in their paper), whereas infants in another group saw a sequence in which successive items were likely to be dissimilar. Importantly the actual stimulus set for both groups was identical—only the order changed. Their results indicated that infants in the 'high distance' (dissimilar successive items) condition were likely to show novelty preference for an atypical item, whereas infants in the 'low distance' (similar successive items) condition were likely to show a null preference. The authors argued that switching between dissimilar items during familiarization allowed infants to mentally represent a greater amount of the feature space (see [45] for discussion). While originating from a hypothesis that was based on computational modelling with self-organizing maps [45], Mather & Plunkett's [46] results were subsequently also simulated with an auto-encoder network by Twomey & Westermann [47], which successfully captured the difference between high- and low-distance sequences. The inherent geometry of the feature space used in this experiment appears to explain the variability between infants' looking preferences, and can be captured by a learning device that is geared towards discovering regularities within this space.

Here we shall argue, however, that a more compelling explanation is that these results are driven by memory constraints and priming effects.

# 3. Rationale: hypothesizing recency effects

Our starting point was a simulation of familiarization with sequences used by Mather & Plunkett [46]. We used Gliozzi *et al.*'s [44] model (subjected to small adaptions, see below). This will be presented below in detail as Simulation 1. We expected the model to exhibit a difference between high and low average Euclidean distance sequences, as reported by Mather and Plunkett. The model indeed behaved as expected during the test trials—i.e. it exhibited a higher preference for the peripheral item when trained with a high-distance condition. However, inspecting model behaviour during training showed an unexpected effect: it appeared that a better predictor of test performance was the final familiarization item. The stimulus set used by Mather & Plunkett [46] was originally constructed by Younger [3]. The pair of test stimuli is constructed such that one represents the overall average of all familiarization items, while the other represents an outlier, or peripheral item, in the feature space. As we shall see below, in Simulation 1 we observed that whenever the final familiarization item was close to the overall average of all stimuli, the network ended up exhibiting a clear preference for the peripheral item at test. Whenever that last familiarization item was further away from the overall average, the model's quantization error for the two test items was approximately equal. Since the modelling results in general corresponded remarkably well to the infant data reported by Mather & Plunkett [46] this led us to hypothesize that recency effects were responsible for test performance in infant categorization.

In the following section, we first present the model itself. We then proceed to a formal presentation of Simulation 1, which consists of modelling Mather & Plunkett's infant experiments. Then we set out to test the hypothesis that recency effects are in fact the driving factor behind novelty preference in test trials. Testing this involves a new $2 \times 2$ experimental design with sequences manipulating both the factors Euclidean distance and the type of final familiarization stimulus. We first report simulation results using these new sequences (Simulation 2), which makes clear predictions for infant experiments. These are confirmed by our studies with 10-month-olds, which we finally present in the section 'Infant experiment'.

# 4. Simulations

## 4.1. Model overview

The novelty preference task, and categorization processes, can be simulated using neural network models. Here we use a self-organizing map (or SOM) which emphasizes the process of learning category structure *online* in a training procedure similar to that experienced by infants during familiarization. Training involves presenting each familiarization stimulus exactly once, and the discrepancy between the model's internal category representation and the currently presented stimulus represents looking time. The self-organizing map consists of a two-dimensional layer of interconnected units, fully connected to an input vector that represents successive training stimuli. Each unit is associated with a position in feature space, and the model responds to a new stimulus by activating the unit that is closest to the stimulus's feature representation. To determine this, we calculate the 'quantization error', or distance of

each unit in the map to the input. The 'best-matching unit (BMU)' is the one with the lowest error. Learning involves moving this unit even closer to the stimulus representation (reducing the error), and also moving other units close by in the same direction. The quantization error associated with the BMU in the model is the equivalent of infant looking time. In our model, learning differs from standard self-organizing maps in three ways. Firstly, the learning rate is higher than usual because rather than presenting the entire stimulus set hundreds of times and learning very gradually, we aimed to simulate learning exactly as infants do, i.e. presenting each stimulus exactly once. Secondly, the learning rate is adaptive (as in [44]) and changes with the quantization error: the more novel a stimulus is (higher quantization error) the higher the learning rate becomes. In other words, a surprising stimulus will affect the learning rate to a larger degree than an expected one. This is in line with the finding that infants' looking time during familiarization is higher for more novel items, and this is interpreted as the time taken to incorporate that stimulus into the category representation [43,48]. Thirdly, we consider a simplified case of self-organizing maps in which only the BMU is involved in learning. In standard self-organizing maps learning affects both the best-matching unit and its neighbouring units, which are also moved closer to the stimulus representation. In our map only the BMU itself is moved, meaning the map does not necessarily form a topological representation of the input space.

Assessing preferential looking after familiarization with a set of stimuli then implies measuring the quantization error for each of the test stimuli. Again, quantization error is taken to correspond to infant looking time. The expectation is that a within-category stimulus will be relatively close to an existing unit, because one or more units were moved into its direction during training as BMUs of familiarization exemplars. In other words, a within-category stimulus might share its BMU with one or more familiarization stimuli. The result is that the within-category stimulus will not generate a large quantization error. An OOC item, by contrast, will generate high error (i.e. be far away from the best-matching unit) because no unit has been moved in that direction. A 'preference score' is then calculated by directly comparing quantization error for the two items shown during a test trial. A novelty preference is found if the quantization error is larger for an OOC item, whereas a null preference is found if the quantization error is approximately the same for both test items.

SOMs implement a biologically plausible approach to human information processing that has been applied to a wide range of cognitive phenomena (see [49] for a review). The big advantage of such a model is that it allows us not just to probe the response to a given test stimulus after training. We can also ask questions about the nature of the existing representations the model has learned, i.e. the internal configuration of units and their distance to stimuli.

In our simulations, we used Younger's [3] stimulus set. Previous approaches to simulating infant familiarization with these images (e.g. [47,50,51]) have used auto-encoder networks using backpropagation [52]. While this is similar to using SOMs in the sense that infant looking time in both models is represented by classification error, the approaches differ in terms of how the models are trained, and what the internal representation looks like.

Both approaches are capable of capturing sequence effects such as those demonstrated in Mather & Plunkett's data (see [47] for a simulation of that dataset using autoencoders). However, the mapping of individual category exemplars to identical versus different BMUs, which provides immediate insight into the internal category representation that is formed, make SOMs particularly intuitive to use in our investigation.

### 4.1.1. Stimuli

The stimuli consist of 12 line drawings (figures 1 and 2) of novel animals—the so-called Broad condition and test stimuli in [3]. Stimuli are defined by four features: length of the legs, length of the neck, tail width and distance between the ears, which each can take five values (1–5). This means that 3333 represents the 'overall average' stimulus. This item is not shown during familiarization, but used as a test item. 5555 and 1111, the so-called 'peripheral' stimuli, which are as far as possible from this overall average, are used as the remaining test items. Just like 3333, these items are not presented during familiarization.

Particularly relevant is the aspect that all familiarization stimuli consist of either features with values 1 and 5, or 2 and 4 (but not mixed). This means that each item is either relatively close to the overall average (items with values 2 and 4) or quite far from the overall average (items with values 1 and 5). We will refer to these as **'near'** versus **'far'** stimuli (for illustration, see figure 3).

The logic of the design is this: since infants are exposed to items with values 2/4, 1/5 during familiarization, they are expected to form a category with a central tendency around 3333. Being presented with 3333 versus 5555 or 1111 (peripheral items) on the test trials should then result in a

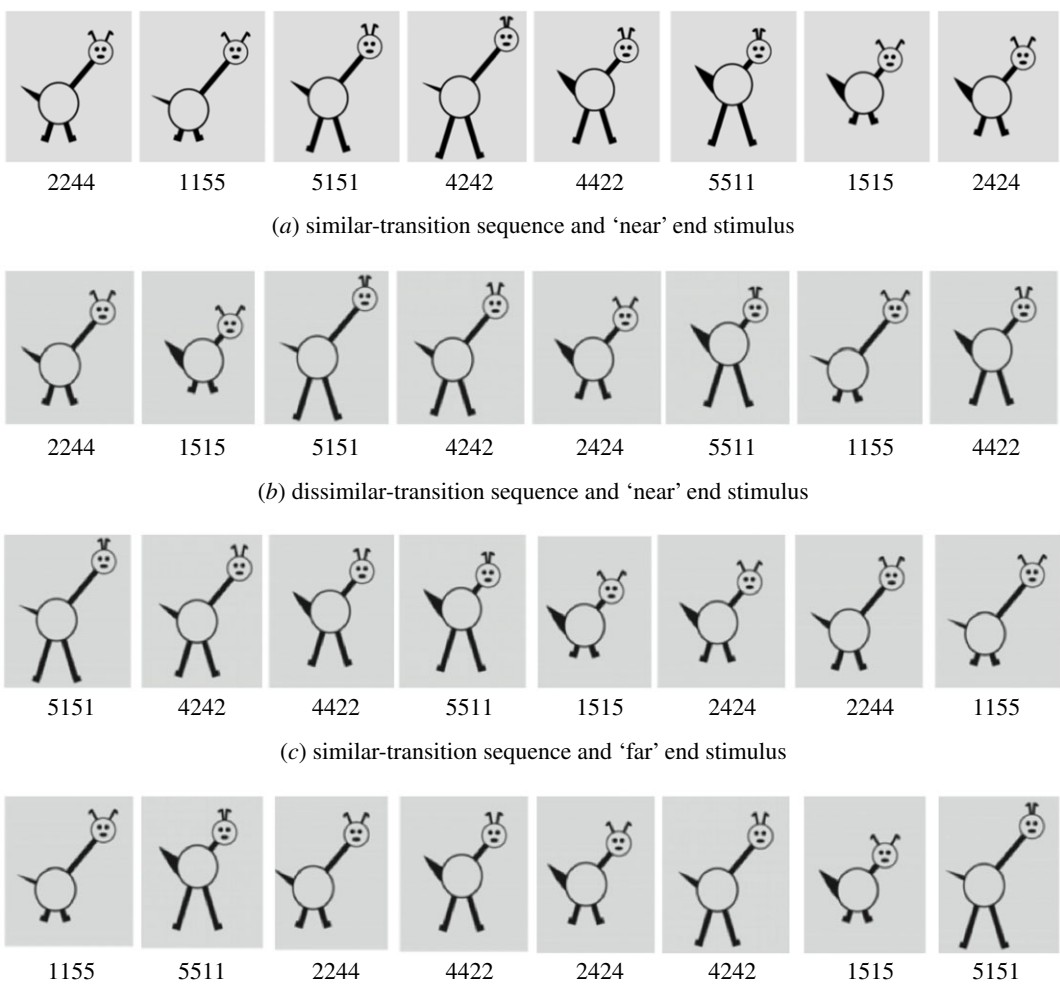

**Figure 1.** Examples of familiarization sequences in the four conditions. Note that each sequence contains the same set of eight objects. (*a*) Similar-transition sequence and 'near' end stimulus, (*b*) dissimilar-transition sequence and 'near end' stimulus, (*c*) similar-transition sequence and 'far end' stimulus, (*d*) dissimilar-transition sequence and 'far end' stimulus.

preference for the peripheral item, as 3333 ought to appear relatively more familiar by this stage. Younger [3], as well as Plunkett *et al.* [7] reported results consistent with this prediction.

It is also important to note that the exact sequence of stimuli during familiarization is relevant for learning. Mather & Plunkett [46] exposed infants to sequences either involving mainly transitions between highly similar items (e.g. from 2244 to 2424), or involving mainly transitions between highly dissimilar items (e.g. from 1155 to 5511). These were termed 'low Euclidean distance sequences' and 'high Euclidean distance' sequences, respectively. Infants in the high distance (dissimilar transitions) condition were more likely to show a preference for the peripheral test item than infants who saw a sequence containing many transitions to similar stimuli, despite using the same set of stimuli in each sequence. Here, we use a corresponding sequence manipulation in our simulations and experiments, with the aim of dissociating the sequence effects from recency effects. In order to distinguish the sequence manipulation from the other experimental manipulation (the type of final familiarization item) we shall refer to the sequence types as **similar-transition sequences** (low Euclidean distance in [46]) versus **dissimilar-transition sequences** (high Euclidean distance in [46]). The final familiarization stimulus types, by contrast, will be referred to as **near** (containing values 2/4; therefore nearer to the overall average 3333) and **far** (containing values 1/5; therefore further away from the overall average 3333).

In order to train self-organizing maps, the drawings were encoded as four-dimensional vectors (each value corresponding to one feature). These were obtained as described by Mareschal & French [51], see table 1.

### 4.1.2. Formal model specification

Our model is a self-organizing map (SOM), schematically depicted in figure 4.

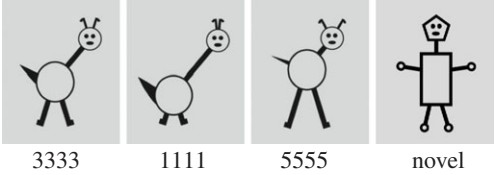

**Figure 2.** Test stimuli: Average (3333), peripheral stimuli (1111 or 5555) and novel OOC item.

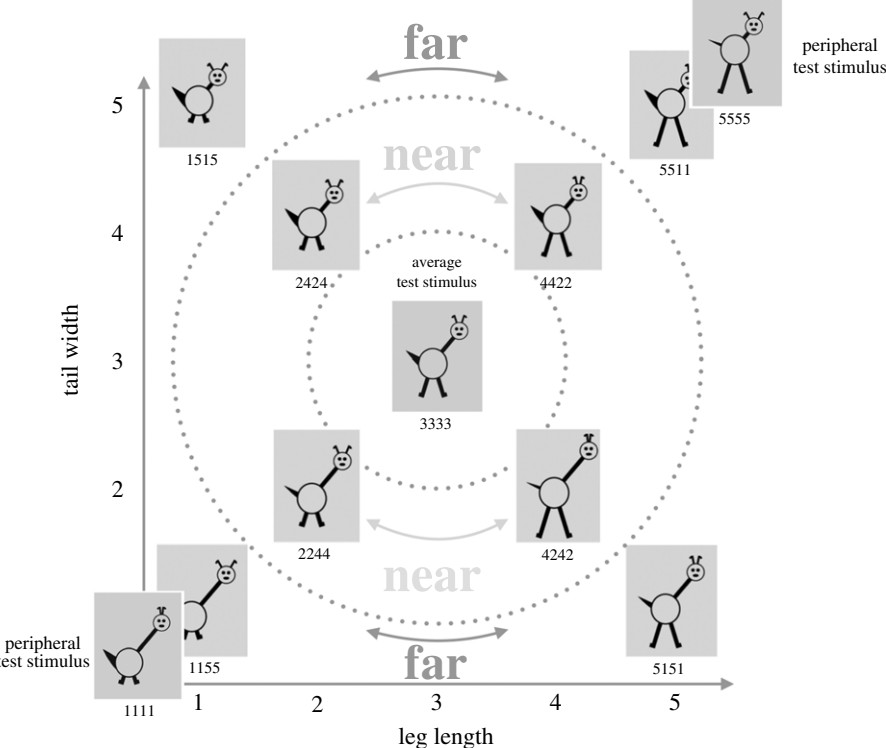

**Figure 3.** Illustration of the feature space, using just the first two features to allow a two-dimensional representation. 'Near' stimuli are closer to the overall average (3333) in terms of Euclidean distance; 'far' stimuli are further away.

A SOM is a set of units, spatially organized in a rectangular grid. We use a map with nine units, as in [44]. Each map unit $u$ is associated with a weight vector $W_u$ of the same dimension of the input vectors (in this case of dimension 4). We will sometimes call $W_u$ the 'representation' associated with the unit $u$. All weight vectors taken together can be seen as the map's representation of the world. At the beginning, they are initialized to very small random values. During training, the input vectors are sequentially presented to the network. After each presentation, the BMU is identified as the unit whose weight vector is closest to the input vector itself (in Euclidean distance). Next, weights are adjusted to decrease the difference between the current input vector and the weight vector associated with the BMU according to equation (4.1)

$$W_u(t+1) = W_u(t) + a(t) * h_{u,BMU}(t)(I(t) - W_u(t)), \tag{4.1}$$

where $W_u(t+1)$ and $W_u(t)$ are the weight vectors associated with unit $u$ at time $t+1$ and $t$, respectively; $a(t)$ is the learning rate (see discussion below); $h$ is the neighbourhood function and $h_{u,BMU}(t)$ indicates the distance between $u$ and BMU at time $t$: in our model the neighbourhood function is the Kronecker function (i.e. for all $t$, $h_{u,BMU}(t) = 1$ if $u = BMU$, 0 otherwise). This is different from standard self-organizing maps in which the neighbourhood function is a Gaussian involving several units of the map. $I(t)$ is the input vector presented to the network at time $t$. For the best-matching unit $BMU$ and for input $I(t)$, the Euclidean distance between $I(t)$ and $W_{BMU}(t)$ is called the *quantization error* (qerr for short) of the network with respect to $I(t)$.

$$qerr = \|I(t) - W_{BMU}(t)\|. \tag{4.2}$$

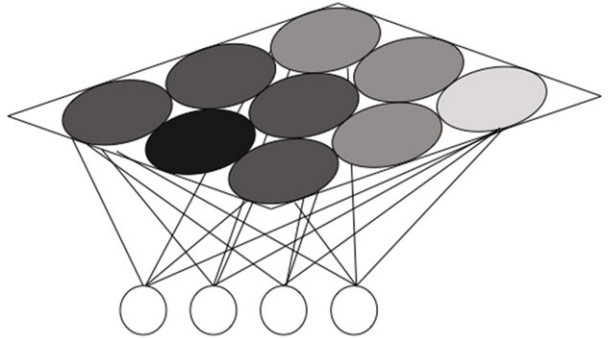

**Figure 4.** The self-organizing map.

**Table 1.** Stimulus encoding for the Younger [3] stimuli.

| legs | neck | tail | ears |
| --- | --- | --- | --- |
| 0.27 | 1.00 | 0.22 | 1.00 |
| 0.27 | 0.23 | 1.00 | 1.00 |
| 0.45 | 0.81 | 0.41 | 0.78 |
| 0.45 | 0.42 | 0.80 | 0.78 |
| 0.82 | 0.42 | 0.80 | 0.33 |
| 0.82 | 0.81 | 0.41 | 0.33 |
| 1.00 | 0.23 | 1.00 | 0.11 |
| 1.00 | 1.00 | 0.22 | 0.11 |

The main differences between the current model and standard SOMs are the simplified neighbourhood function and an adaptive learning rate $a(t)$ which, unlike standard SOMs, depends on the quantization error (the higher the quantization error, the higher the learning rate).

The learning rate is defined as:

$$a = \max(0, \min(1, \beta * \exp^{\alpha * \sqrt{\text{qerr}}})). \tag{4.3}$$

For the simulations presented here, the parameters are set to $\alpha = 1.8$, $\beta = 0.4$. The model's learning rate has two important properties. First, it is typically higher than in standard SOMs. This permits the network to be trained in an analogous fashion to the infant familiarization procedure: presenting each familiarization stimulus to the network just once, rather than training the network over hundreds of epochs, effectively presenting each stimulus many times. Second, because the learning rate depends on the quantization error, the more novel the incoming stimulus, the higher the learning rate will be. In this way, the learning rate is related to attention in infants: the adaptive learning rate corresponds to the general finding that infants pay more attention to novel stimuli rather than to familiar ones. This is consistent with theoretical and computational models of habituation such as the comparator theory [48,53,54].

## 4.2. Simulation 1: preliminary simulations to model sequence effects

In order to simulate the infant results reported by Mather & Plunkett [46], we trained 24 SOMs (corresponding to 24 infants) in the two conditions containing original similar-transition sequences and dissimilar-transition sequences (low versus high Euclidean distance sequences in their terminology). Each simulation involved one stimulus sequence used in their original work.

After each network was trained, we assessed category formation by measuring *network looking time*, i.e. the quantization error associated with the BMU (see above), at the test stimuli. As with the infant experiments, a preference score was calculated by dividing network looking time at the peripheral test

stimulus by the overall network looking time at the test stimuli (network looking time at the average test stimulus + network looking time at the peripheral test stimulus).

For each condition, the average of these preference scores for all networks was calculated, and compared to the corresponding preference scores presented by Mather & Plunkett [46]. The model reproduced Mather & Plunkett's [46] results with infants: networks familiarized with dissimilar-transition sequences exhibited a stronger novelty preference for the peripheral test stimulus than those familiarized with similar-transition sequences. The average preference score for similar-transition sequences was 0.59, whereas in the dissimilar-transition sequences it was 0.66. A two-tailed $t$-test indicated this difference was significant ($t_{46} = 60$, $p < 0.001$). Results were robust (i.e. held in more than 75% of the cases) when the learning rate parameter $\alpha$ was in the range from 0.1 to 4, and $\beta$ ranged from 0.1 to 1.

However, we further observed in these simulations that another explanation for differences between the conditions could be the final familiarization stimulus. Since sequences in [46] were optimized for minimal/maximal transition similarity, the sequences in their dissimilar-transition sequences (high Euclidean distance) ended in 'near' stimuli, and those in similar-transition sequences (low Euclidean distance) ended in 'far' items. This was particularly predictive of the behaviour at test—there seemed to be a clear recency effect in the dataset. However, using these existing sequences, it is impossible to evaluate the relative contributions of Euclidean distance (transition similarity) and recency effects. For this reason, we created a new set of sequences, crossing the factors transition similarity and recency.

## 4.3. Simulation 2: main simulations of recency effects

To test the hypothesis that preferential looking at test is driven primarily by recency effects we set up a new set of simulations. The aim was (i) to understand the mathematical underpinnings of the recency effects, and (ii) to provide a testable prediction for a further set of infant studies. In order to evaluate the hypothesis that transition similarity (average Euclidean distance between successive stimuli) is the most important factor, as maintained by Mather & Plunkett [46], we constructed the simulations in a $2 \times 2$ design with factors transition similarity and end stimulus type.

This involved the following combinations: similar-transition sequence and 'near' end stimulus; similar-transition sequence and 'far' end stimulus; dissimilar-transition sequence and 'near' end stimulus; dissimilar-transition sequence and 'far' end stimulus. Figure 1 shows example sequences. We selected sequences on the basis of having extremely low or extremely high average transition similarity (in order to maximize the potential difference of this variable, cf. [46]) and ending with near or far stimuli, keeping the average transition similarity equal across end stimulus type.[1]

On the basis that infant studies using this stimulus set typically included 24 infants per condition [3,7,46], we trained 24 SOMs in each condition (with 24 different sequences, just as 24 infants are typically exposed to 24 different sequences, see below in 'Infant experiment'). We then tested whether the SOM exhibited a preference for the average or peripheral test stimulus, by presenting each stimulus in turn (3333 or 1111/5555) and calculating the corresponding quantization errors.

### 4.3.1. Results

These simulations revealed a much more pronounced preference for the peripheral item after training with sequences ending in near stimuli compared to sequences ending in far stimuli. In other words, the model predicts a recency effect. The results, depicted in figure 5, show the proportion of network looking time directed at the peripheral test stimulus out of the total network looking time for the four conditions.[2]

For these simulations, an ANOVA with factors transition similarity (similar-transition sequence versus dissimilar-transition sequence) and end stimulus type (near versus far) revealed a main effect of end stimulus type ($F_{1,92} = 65.73$, $p < 0.001$), no main effect of transition similarity ($F_{1,92} = 0.39$, $p = 0.5$), and a trend for an interaction of transition similarity × end stimulus: ($F_{1,92} = 3.0$, $p = 0.08$).

[1]A side effect of minimizing/maximizing transition similarity is that the first stimulus in the sequence is from the same category (near or far) as the final stimulus. The implications of this constraint are discussed below.

[2]Note that the lack of variability we observe in the model results is a direct consequence of three factors: (i) the high learning rate, (ii) no noise being introduced in the stimulus encoding, and (iii) training involving only eight weight updates.

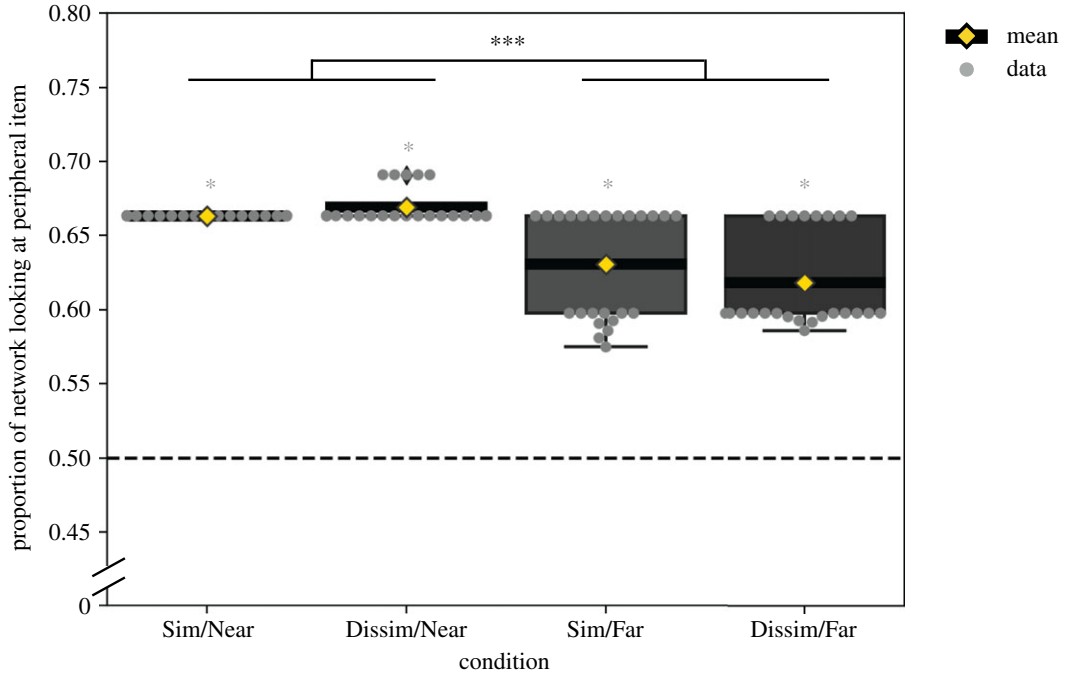

**Figure 5.** Model preference scores for the peripheral stimulus in test trial 1. Boxes show quartiles, whiskers show the range. Grey horizontal lines show medians. Grey dots show individual simulation results. Yellow diamonds show the means. The significant ANOVA main effect is indicated by black asterisks (***$p < 0.001$), significant single-sample $t$-tests against chance are indicated by grey asterisks (*$p < 0.05$). Sim, similar-transition sequence; Dissim, dissimilar-transition sequence; Near, end stimulus close to overall average; Far, end stimulus distant from overall average.

The model's prediction of a recency effect (end stimulus type) is robust, holding in more than 75% of the cases in the parameter range considered. If the final familiarization stimulus is 'near', the network predicts a stronger novelty preference than if the final stimulus is 'far'.

In order to assess representations in the SOMs and gain further insight into the learning mechanism, specifically with regard to the question of why the model exhibits recency effects, we determined the number of *different* BMUs activated per simulation. This showed that 12% of simulations used four different BMUs, 45% used three different units, 43% of simulations used two, and no simulations relied on a single BMU. The number of different BMUs was further not the same across conditions. An ANOVA with factors transition similarity and end stimulus type revealed a main effect of the end stimulus type ($F_{1,92} = 5.81$, $p = 0.018$), a main effect of transition similarity ($F_{1,92} = 7.46$, $p = 0.008$), and an interaction of transition similarity × end stimulus type ($F_{1,92} = 4.36$, $p = 0.040$). *Post hoc* tests showed that sequences with a 'near' end stimulus and low transition similarity typically used more different BMUs ($M = 3.17$, med = 3, s.e. = 0.115, $p < 0.05$) than all other sequences (similar/far: $M = 2.58$, med = 3, s.e. = 0.119; dissimilar/near: $M = 2.54$, med = 2, s.e. = 0.134; dissimilar/far: $M = 2.50$, med = 2, s.e. = 0.147).

## 4.4. Discussion

The result of our simulations clearly shows that the end stimulus type is the dominant factor in determining responses at test. The preference for the peripheral test stimulus is much stronger for sequences ending in 'near' items than for those sequences ending in 'far' items. In other words, the model's behaviour is completely consistent with a recency effect, whereas the effect of the average transition similarity across the whole of the familiarization phase (as indexed by the trend for an interaction) plays only a minor role.

How does this come about? Inspecting internal representations in the SOMs allows us to gain more insight. Initially, a BMU is selected for the first observed stimulus, and its vector representation is updated accordingly. For subsequently presented stimuli, this BMU is either updated to match the incoming stimulus or, if there is a different, better-matching unit, that other unit is chosen as the BMU. Due to the high learning rate, at every point in time the item that is likely to cause lowest error

in the model is always the most recently presented exemplar (as individual unit vectors have just been adjusted to match that item).

At the time the test item is reached, similarity between the previous item and the test item therefore matters a lot. If the most recent item was a 'near' exemplar, its BMU will have been adjusted towards the 'near' feature values. It is therefore likely that the model's BMU for the average test stimulus (3333) will be that same unit, and quantization error will be low. For the peripheral test stimulus (1111/5555) another BMU will most likely selected and its quantization error will be higher than the average test stimulus's one (even if the *same* unit is selected as best-matching, it cannot match as well as for the average, and necessarily produces higher quantization error).

The inverse holds for training sequences ending in 'far' items: because the BMU for that last item was just moved towards those far feature values, the existence of a unit with feature values close to the average is no longer guaranteed. For this reason, the quantization error for the two test items is likely to be quite similar, with no preference emerging. Geometrically speaking, with a sequence ending in a 'near' stimulus, the current category representation is close to the overall average. With a sequence ending in a 'far' stimulus, the category representation lies between the overall average and the peripheral stimulus.[3]

The difference between sequences also had an effect on the internal representation of the categories in terms of the number of different BMUs selected during training. Dissimilar-transition sequences ending in a 'near' item led to a more distributed representation involving more BMUs, compared to the other sequence types. The internal representation of categories therefore seems to depend on both factors, transition similarity and end stimulus type.

Partially responsible for the recency effect is the adaptive learning rate: If the learning rate is sufficiently high, the update overrides previous representations, and as a result the last experienced stimulus will be the best represented one. Note that this adaptive learning rate was not built into the model explicitly to cause a recency effect, but merely to incorporate what we know about the interplay of relative novelty of an item, and subsequently increased looking time (cf. [44]).

The model offers a clear set of predictions regarding the conditions under which infants should exhibit a preference for the peripheral test stimulus when familiarization sequences are carefully controlled for transition similarity and final familiarization stimulus type. In particular, the model predicts that this preference should be enhanced when the final familiarization item is 'near'.

In the following section, we present data from 10-month-old infants, familiarized with the same sequences as the models in this section, in order to put this prediction to the test.

# 5. Infant experiment: testing recency effects in 10-month-olds

## 5.1. Methods

### 5.1.1. Participants

In total, 96 infants (mean age: 311 days (range: 289–329 days); 47 females) took part in this study. Aiming for 24 infants in each condition, as in previous work, we recruited 135 infants in total to allow for the typical drop-out rate. Thirty-nine infants were excluded due to technical reasons ($N = 12$) or a failure to reach looking time criteria ($N = 27$). The looking time criteria were as follows: infants were excluded if they missed three or more trials during familiarization or if their total looking time was lower than 2 standard deviations below the mean total familiarization time. These criteria were applied in order to ensure that the set of infants we are comparing at test had similar amounts of exposure to the target category. Infants were further excluded if they missed trial 1 or 8 or test trial 1, as these were critical to our design. The remaining 96 infants were allocated to one of four conditions, similar-transition sequence and 'near' end stimulus ($N = 26$), similar-transition sequence and 'far' end stimulus ($N = 22$), dissimilar-transition sequence and 'near' end stimulus ($N = 25$) and dissimilar-transition sequence and 'far' end stimulus ($N = 23$). Infants were previously recruited from the maternity ward of the local hospital and had no known auditory or visual impairments.

---

[3]Due to the nature of the feature distribution, the category representation can mathematically never be quite so close to the peripheral item as to the overall average. For this reason, in absolute terms, we always see a preference for the peripheral item (preference scores > 0.5).

### 5.1.2. Stimuli and design

Infants were familiarized with stimulus sequences corresponding to those used to train the model. As explained above, these came in four conditions which differed with regard to transition similarity (similar/dissimilar) and were further constrained to contain either 'near' (feature values 2/4) or 'far' (feature values 1/5) end stimuli.

The test stimuli were the overall average stimulus (3333), two peripheral stimuli (1111/5555), and an OOC object and were identical for all four conditions (figure 2).

### 5.1.3. Procedure

After arrival at the laboratory, written consent was obtained from carers and the procedure was explained by the experimenter. Infants were allowed to play with some toys during this period which lasted no longer than 10 min. Infants were then seated on the carer's lap in front of a large television screen (110 × 95 cm) at a distance of approximately 90 cm. The screen was located in a booth with dark grey walls, and a curtain was closed behind the carer's chair. The carer was instructed to keep their eyes closed and to not interact with the infant or interfere with their looking. The carer was also wearing headphones and listening to music. For the duration of the experiment, the experimenter was in a control room adjacent to the testing booth, and watched the infant's face via a camera system.

After two animations that served to direct infants' attention to the centre of the screen infants were presented with eight familiarization trials, followed by four test trials. All trials were 10 s in length. During the eight familiarization trials, a single familiarization image was displayed either on the left- or right-hand side of the screen (at 2.5° visual angle from the centre). During the test trials, two images were shown side-by-side in the same positions as the familiarization stimuli. All images subtended 13.73° visual angle. The first two test trials paired one of the peripheral stimuli with the overall average (either 1111 and 3333, or 5555 and 3333). The location of the average stimulus was switched between the two trials, and its location on the first test trial was counterbalanced across participants. Test trials 3 and 4 involved the OOC stimulus. On one of the two trials this was paired with the overall average (3333) and on the other it was paired with the same peripheral stimulus that appeared in test trials 1 and 2. The order of the two cases was counterbalanced across infants. The location of the OOC stimulus was switched between test trials 3 and 4, and its location on trial 3 was counterbalanced across participants. Note that the model only makes predictions for the first test trial. Test performance in the model is assessed without further updating the model's weights. Infants, on the other hand, continue updating their mental representation after familiarization ends—there is no mechanism to 'fix' the representation after trial 8. Their mental status after the first test trial is therefore not simulated in the model. We included test trials 2–4 in our report of the infant experiment because such trials can provide further information about infants' mental categories. The novelty preference trials (test trials 3 and 4) are necessary in infant studies to determine whether infants are in familiarity preference or novelty preference mode during tests 1 and 2. Trials 2 and 4 are included for counterbalancing purposes and to guard against side biases in individual infants.

Trials were initiated manually by the experimenter after confirming that the infant's gaze was directed at the screen. If the infant was not paying attention to the screen between trials, the experimenter attracted attention to the display by talking to the infant via a microphone, e.g. by saying 'Look (baby's name)! What can you see?' (or similar) in an infant-directed voice. The infant's face was filmed by two cameras mounted above the screen one on the left-hand side and one on the right-hand side.

## 5.2. Results

The video streams from left and right cameras were manually scored frame-by-frame for infants' gaze direction (Intercoder reliability: $r = 0.96$, on approx. 5% of trials independently double-scored).

### 5.2.1. Looking time during familiarization

To evaluate the degree of infant familiarization, we split the learning phase into two blocks of four trials (block 1: trials 1–4; block 2: trials 5–8) in order to compare how much time infants spent investigating a stimulus at the beginning versus the end of familiarization. Average trial looking times were obtained for each infant and each block. Figure 6 depicts the looking time data for each condition and block.

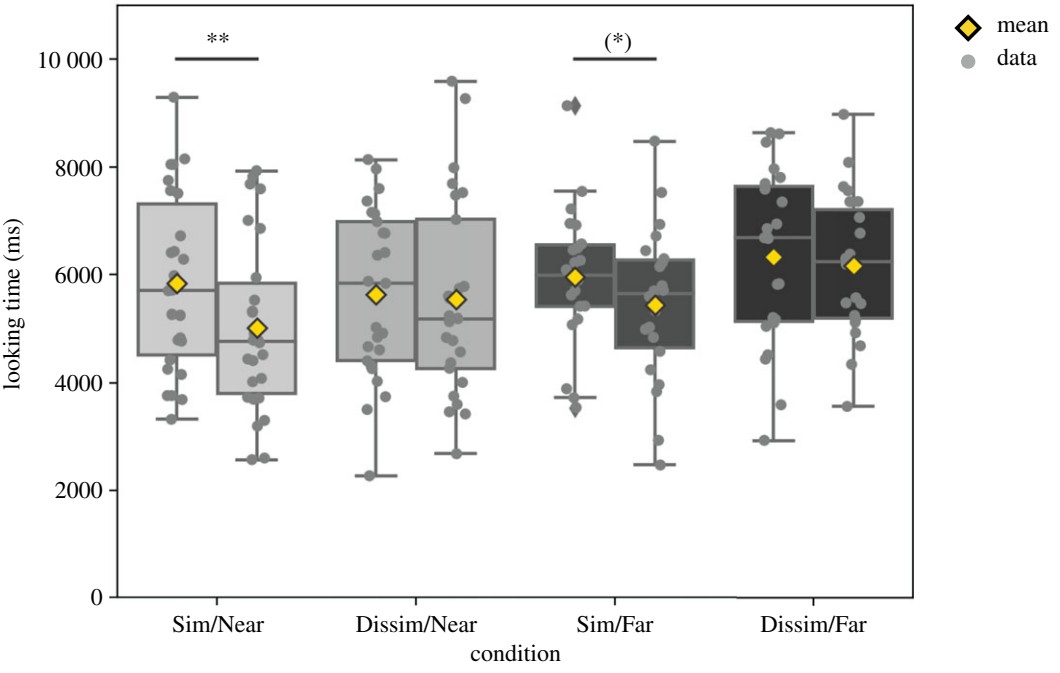

**Figure 6.** Looking time (ms) during familiarization. Boxes show quartiles, whiskers show the range. Grey diamonds indicate outliers. Grey horizontal lines show medians. Grey dots show individual infants' data. Yellow diamonds show the means. Significant pairwise t-test results are indicated by asterisks (\*\*$p < 0.005$, (\*) $p < 0.1$). Sim, similar-transition sequence; Dissim, dissimilar-transition sequence; Near, end stimulus close to overall average; Far, end stimulus distant from overall average.

Overall, looking times were longest for dissimilar-transition sequences with 'far' end stimulus, and shortest in the similar-transition sequences with 'near' end stimulus. These looking time data were subjected to a mixed measures ANOVA with within-subjects factor block (1,2) and between-subjects factors transition similarity (similar versus dissimilar) and end stimulus type (near versus far). This revealed a main effect of block ($F_{1,92} = 7.232$, $p = 0.009$, $\eta_p^2 = 0.073$) and a trend for a block × transition similarity interaction ($F_{1,92} = 3.262$, $p = 0.074$). Paired t-tests confirmed that looking time decreased between blocks 1 and 2 for similar-transition sequences ($t_{47} = 3.667$, $p = 0.001$, $d = 0.53$; individual conditions: similar-transition sequence and 'near' end stimulus: $t_{25} = 3.401$, $p = 0.002$, $d = 0.67$; similar-transition sequence and 'far' end stimulus: $t_{21} = 1.78$, $p = 0.089$, $d = 0.38$), but did not differ systematically for dissimilar-transition sequences ($t_{47} = 0.568$, $p = 0.573$). This indicates that infants seeing similar-transition sequences were familiarized to a greater degree than infants seeing dissimilar-transition sequences. This is consistent with the hypothesis that sequences with transitions to dissimilar items maintain attention better than sequences with transitions to similar items due to the greater novelty of successive stimuli [45].

### 5.2.2. Categorization: test trial 1

Since the model simulates test trial 1 only, this is the most important trial for our purpose.

In order to assess categorization performance, preference scores were obtained from each participant by dividing the time spent looking at the peripheral stimulus, 1111/5555, by the time spent looking at both test stimuli, average and peripheral (see figure 7 for results). The resulting preference scores from the first test trial were subjected to an ANOVA with factors transition similarity (similar versus dissimilar) and end stimulus type (far versus near). This revealed a main effect of end stimulus type ($F_{1,92} = 6.242$, $p = 0.014$, $\eta_p^2 = 0.064$). All other effects remained non-significant (all $Fs < 0.31$, $ps > 0.57$). Follow-up t-tests on collapsed data showed that infants in the 'near' end stimulus conditions exhibited a preference for the peripheral stimulus on test trial 1 (looking proportion for 1111/5555: $M = 0.58$, s.e. $= 0.03$; $t_{50} = 2.882$, $p = 0.006$, $d = 0.4$; a Bayesian one-sample t-test resulted in $BF_{10} = 5.929$, which implies sufficient evidence to reject the null hypothesis), whereas infants in the 'far' end stimulus conditions exhibited no preference (looking proportion for 1111/5555: $M = 0.49$, s.e. $= 0.02$; $t_{44} = 0.564$, $p = 0.576$).[4] The ANOVA results, revealing only a main effect of end stimulus type, confirm the

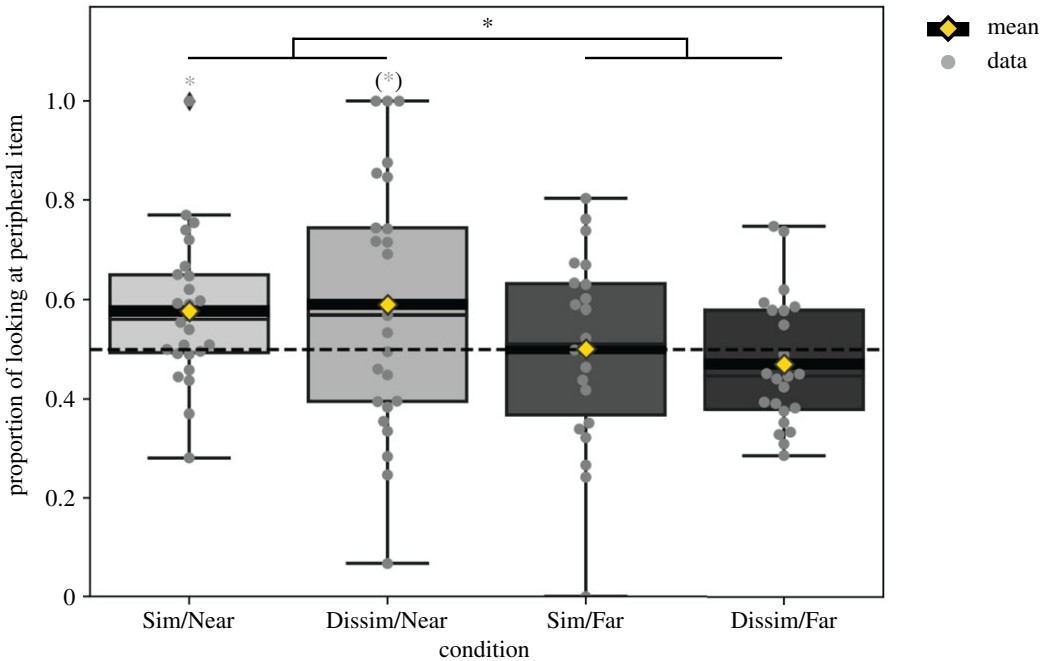

**Figure 7.** Infant preference scores for the peripheral stimulus in test trial 1. Boxes show quartiles, whiskers show the range. Grey diamonds indicate outliers. Grey horizontal lines show medians. Grey dots show individual infants' data. Yellow diamonds on thick black horizontal lines show the means. ANOVA main effects are indicated by black asterisks (*$p < 0.05$), single-sample *t*-test results are indicated by grey asterisks (*$p < 0.05$, (*) $p < 0.1$). Sim, similar-transition sequence; Dissim, dissimilar-transition sequence; Near, final familiarization item close to overall average; Far, final familiarization item distant from overall average.

model's prediction that novelty preference is driven by a recency effect, in particular with effects of transition similarity not modulating results further.

### 5.2.3. Categorization: test trial 2

Although the model makes no predictions for test trial 2, this trial can provide further insight into the infants' behaviour. The observed pattern of looking was clearly different from test trial 1. An ANOVA with factors transition similarity and end stimulus type revealed a significant interaction ($F_{1,90} = 7.309$, $p = 0.008$, $\eta_p^2 = 0.077$). No other effects were significant (all $Fs < 0.74$, $ps > 0.39$). Further analysis of the interaction showed that only infants familiarized with dissimilar-transition sequences that ended on a 'far' stimulus had a significant preference for the peripheral stimulus ($M = 0.6$, s.e. $= 0.2$, $t_{23} = 2.47$, $p = 0.022$, $d = 0.5$). Preferential looking in all other conditions did not differ from chance (0.5; all $ts < 1.2$, $ps > 0.24$; individual conditions: similar-transition sequence/near $M = 0.53$, s.e. $= 0.04$; dissimilar-transition sequence/near $M = 0.44$, s.e. $= 0.05$; similar-transition sequence/far $M = 0.43$, s.e. $= 0.06$).

### 5.2.4. Novelty preference: test trials 3 and 4

In order to establish that looking on the first test trials was driven by novelty preference rather than familiarity preference, preference scores were obtained for test trials 3 and 4 by dividing the amount of looking at the novel stimulus by the total looking time for each trial. An ANOVA with repeated factors trial number (3,4) and fixed factors trial order ('novel versus peripheral' first, 'novel versus average' first), transition similarity, and end stimulus type revealed only a main effect of trial order ($F_{1,81} = 6.38$, $p = 0.013$, $\eta_p^2 = 0.073$). All other effects were non-significant ($Fs < 2.3$, $ps \geq 0.14$). Follow-up tests showed that infants who saw the 'novel versus peripheral' test first exhibited significant novelty preference on both trials (trial 3, novel and peripheral items: $M = 0.68$, s.e. $= 0.036$, $t_{46} = 4.925$, $p < 0.001$,

[4]Single-sample *t*-tests against chance for individual conditions result in the following: similar-transition sequence and 'near' end stimulus $M = 0.58$, s.e. $= 0.03$, $t_{25} = 2.675$, $p = 0.013$, $d = 0.53$; dissimilar-transition sequence and 'near' end stimulus $M = 0.59$, s.e. $= 0.05$, $t_{24} = 1.746$, $p = 0.094$, $d = 0.35$; Bayesian one-sample *t*-test: $BF_{10} = 0.784$, implying insufficient evidence to accept/reject the null hypothesis; similar-transition sequence and 'far' end stimulus $M = 0.50$, s.e. $= 0.04$, $t_{21} = 0.039$, $p > 0.96$; dissimilar-transition sequence and 'far' end stimulus $M = 0.47$, s.e. $= 0.03$, $t_{22} = 1.05$, $p > 0.30$.

$d = 0.72$; trial 4, novel and average item: $M = 0.58$, $t_{43} = 2.354$, $p = 0.023$, $d = 0.28$). Infants who were presented with the 'novel versus average' item first did not reach significant levels of novelty preference but showed trends on both trials (trial 3, novel versus average item: $M = 0.56$, s.e. = 0.03, $t_{46} = 1.82$, $p = 0.075$, $d = 0.27$; trial 4, novel versus peripheral item: $M = 0.55$, s.e. = 0.03, $t_{45} = 1.723$, $p = 0.092$, $d = 0.28$).

## 5.3. Discussion of experimental findings

Test trial 1 is the critical trial in terms of testing the model, since the simulation only generates predictions for one test immediately after familiarization.

The main effect of end stimulus type found for test trial 1 in the infant experiments indicates that a recency effect is the strongest factor, just as predicted by the model. As expected, infants who saw a 'near' stimulus on familiarization trial 8 exhibited a preference for the peripheral stimulus on test trial 1. For these groups, the average stimulus appeared relatively familiar at the time they were presented with test trial 1. Infants in the groups with a 'far' stimulus on trial 8 on the other hand did not exhibit any preference on test trial 1. In other words, we find a recency effect. The peripheral stimulus appears, after the 'near' stimulus on trial 8, as much more surprising than the overall average. Infants in the groups with a 'far' stimulus on trial 8 on the other hand did not exhibit any preference on test trial 1. Empirically, this is surprising since the conventional prediction, according to which sequence does not matter, would be that infants should exhibit a preference for the peripheral stimulus, just as reported by Younger [3]. If we only consider the infant results obtained here, it is difficult to say whether children exposed to sequences ending in a 'far' item learned at all—a null preference is consistent with not having formed a category. However, we also know that the model—which clearly 'categorizes' successfully—predicts a main effect, and in particular a shift of looking towards a null preference. It therefore seems plausible that, rather than not forming a category at all, infants successfully formed a category in the 'far' conditions—one whose centroid is simply located closer to the peripheral stimulus. In fact, according to the model the category representation should be located at a point in between the overall average and the peripheral test item, which mathematically makes both items approximately equally interesting. While the modelling result does not provide conclusive evidence that this is indeed reflective of processes in infants, this is nevertheless a plausible explanation.

Regardless of whether or not infants successfully formed a category in the 'far' conditions, we can conclude that recency effects are important in infant categorization. This is in line with a view of category formation as a fluid, moment-to-moment process that does not result in clear boundaries but representations that are biased by recent experiences: looking time is dependent on priming effects, and this is the consequence of a system that is malleable and adaptive to context.

What our approach does highlight, therefore, is the necessity to take the familiarization sequence (and in particular the final item) seriously as a factor in interpreting looking time. We believe, in fact, that doing so can in some circumstances prevent null results from emerging because different sequences produce systematically different results. The present results provide just such an example: had we collapsed the data from the 'near' and 'far' conditions, we would have obtained an overall non-significant difference between dissimilar- and similar-transition conditions ($t_{94} = 0.226$, $p = 0.821$), with single sample t-tests indicating that preferences in dissimilar-transition sequences did not differ from chance ($M = 0.53$, s.e. = 0.03, $t_{47} = 1.08$, $p = 0.29$), and a trend in the similar-transition sequences ($M = 0.54$, s.e. = 0.02, $t_{47} = 1.7$, $p = 0.097$; $BF_{10} = 0.586$, implying insufficient evidence to either accept or reject the null hypothesis).[5] As our results above show, infants' responses by contrast vary systematically depending on the final familiarization item, and indeed there is a significant preference for those sequences ending with a 'near' item.

Returning to our findings, one question begs to be answered: did infants simply ignore the first seven familiarization items? In other words, is it possible that test behaviour is solely a response to the last item, rather than a response to a set of items? While test trial 1 shows a clear divide between 'near' and 'far' conditions but no role of the familiarization sequence, test trial 2 reveals that infants are not solely responding to that single item. If they ignored the first seven items, then both 'near' and both 'far' conditions should show identical results even on test trial 2. By contrast, infants in the 'far' condition who saw a dissimilar-transition sequence show a preference for the peripheral item, and those who saw a similar-transition sequence continue to exhibit a null preference. Any difference between two

---

[5]Note that this experiment does not use identical sequences to those used in [46], since we maximized/minimized overall Euclidean distance while also holding it stable across sequences with near/'far' end stimuli.

conditions with the same type of end stimulus on familiarization is evidence that infants are processing, not ignoring, the first seven items. The behaviour shown here is further consistent with the idea that infants in dissimilar-transition sequences continue to learn, whereas the more homogeneous similar-transition sequences lead infants to be familiarized to a greater degree.

One further aspect of familiarization needs to be addressed: to test the competing factors of transition similarity versus end stimulus, we selected sequences that minimized/maximized the Euclidean distance. However, due to combinatorics this necessarily results in sequences that either start and end in a 'near' stimulus, or that start and end in a 'far' stimulus. How can we rule out that in fact a primacy effect caused the differences we observed? We suggest that if such were the case infants' category representation should be very settled (rather than fluid) by the time they reach test trials. Test trial 2 should, then, have been very similar to test trial 1, or at any rate the two 'near' and the two 'far' conditions should have resulted in similar patterns, respectively, which they did not (for the 'far' conditions). Consequently, we rule out that our results are driven by primacy effects.

# 6. General discussion

In this paper, we have presented a view of infant categorization under which representations are fluid and adaptable, rather than static, and in which memory plays a major role. Preferential looking is a response to the input encountered on a moment-to-moment basis. As a direct consequence of this, the last item presented during a familiarization phase becomes the strongest factor determining behaviour at test. Our computational model incorporates this behaviour by using an adaptive learning rate that responds to novelty, and ensures that the most recent experiences are reflected most strongly in the category representation. Recency effects are observed in the model as the outcome of this adaptive behaviour.

We presented a set of simulations that provided precise predictions of infant behaviour in a tightly controlled familiarization study where mathematical similarity properties of the stimuli are known. In particular, we pitched the average similarity between successive items—a factor previously thought to determine learning outcomes—against recency effects, or the impact of the final item in the familiarization sequence. Both in the model predictions and in the subsequent infant study using the same familiarization sequences, recency effects dominated test performance. Familiarization with a sequence ending in a 'near' stimulus, i.e. one that was close to the overall average of the stimulus set, led to a preference for a peripheral test item (over the average item). Familiarization with a sequence ending in a 'far' item, by contrast, resulted in a null preference in the infants. This main effect of end stimulus type was present in both model simulations and infants.

Inspecting the model's internal representation further suggested that the main effect, i.e. a shift towards a null preference in the 'far' conditions, may be the consequence of a category representation whose central tendency is between the two test stimuli, rather than reflecting a failure to learn. The model learns in all cases, but its internal representation at test time is biased by the immediately preceding stimulus, such that sequences ending in 'far' stimuli necessarily lead to a representation for which both types of test stimulus are approximately equally interesting.

Our results, demonstrating that the final familiarization item carries considerable importance in determining test outcome, have implications for experimental design. The identity of the final familiarization item should form a part of the experimental design, rather than being factored out by randomizing stimuli. We argue here that this may obscure systematic patterns in infants' performance, in particular when null preferences are obtained. We demonstrated that our set of data would only have resulted in trends without taking this factor into account. Rather than representing a failure to learn, this is instead the consequence of highly systematic behaviour in response to our (specifically selected) stimulus sequences.

The model and data we presented here show that categorization behaviour in infancy is a continuous, dynamic process. The representation of a category is updated on a moment-to-moment basis and biased by most recent experiences. In other words, behaviour on any one trial is primed by the previously seen exemplar. Recency effects as demonstrated here are not a failure of the system to extract an unbiased central tendency of the stimuli. They are the output of a system that is highly adaptive and context-sensitive. These very properties allow cognition to be flexible and create representations that are useful in any given context. The most recent experiences, still represented in memory, are weighted most strongly. Directly observed contrasts, not just general novelty, are therefore emphasized. In our model, representations are constantly adjusted to accommodate new impressions and the 10-month-olds in our study behaved exactly as one would predict if they incorporated such a mechanism.

Finally, our infant data were obtained on the basis of predictions from our model. However, the fact that we obtained the predicted results is no guarantee that model and infants learn in similar ways. Finding further parallels between model and infant behaviour would be a strong argument for the model's validity.

In conclusion, we propose that infant categorization needs to be viewed as context-driven and dynamic, and as such inherently anchored in memory processes. Recency effects are an inherent property of a system that supports flexible adaptation to environments and contexts.

Ethics. The work in this experiment was carried out in accordance with The Code of Ethics of the World Medical Association (Declaration of Helsinki). The work was approved by the University of Oxford's Medical Sciences Division Ethics Committee under the project 'How do infants build a semantic system?' which followed protocol MSD/IDREC/2008/P11.1.

Data accessibility. Model code and infant data can be accessed at the Open Science Foundation at https://osf.io/zt3fd [55].

Authors' contributions. N.A. carried out infant data collection and statistical analysis of infant data. V.G. conducted initial simulations and defined the problem, conducted the self-organizing map simulations and statistical analyses related to them. N.A., V.G., J.M. and K.P. designed the study and wrote the manuscript.

Competing interests. We declare we have no competing interests.

Funding. We received no funding for this study.

Acknowledgements. We are grateful to Linda Smith for helpful comments on an earlier version of this manuscript, and to Katie Twomey and one anonymous reviewer for their thoughtful comments and suggestions. Finally, we wish to thank all parents and infants who participated in this study.

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
