## [Reviewer comments · Royal Society Open Science]

Review History

RSOS-200328.R0 (Original submission)

Review form: Reviewer 1

Is the manuscript scientifically sound in its present form?

Yes

Are the interpretations and conclusions justified by the results?

Yes

Is the language acceptable?

Yes

Do you have any ethical concerns with this paper?

No

Have you any concerns about statistical analyses in this paper?

Yes

Recommendation?

Accept with minor revision (please list in comments)

Comments to the Author(s)

The manuscript describes a computational model and an empirical study investigating categorisation in infancy. Using a SOM-based model, the authors first capture data from an existing empirical study (Mather & Plunkett, 2010), which demonstrated that the order in which 10-month-old infants encounter category exemplars during familiarisation affects their performance at test. Next, the authors used the model to demonstrate that, rather than forming a category representation in which the influence of all exemplars is equal, due to the model's adaptive learning rate the representation is most heavily influenced by the final familiarisation exemplar, resulting in recency effects in test performance. The authors capture this prediction in 10-month-old infants, concluding that infant categorisation is a dynamic, online process, and discussing implications for study design and interpretation.

Overall the paper deals with an important topic of current interest in the field. It will make a strong contribution in terms of theories of infants' online learning and offers a novel computational mechanism for modelling these processes. The model predictions were tested empirically, which is the gold standard in the computational modelling subfield of psychology, and taken together the model and empirical data have both theoretical and methodological implications. Below I have responded to the specific points in the reviewer guidelines for this journal, followed by some questions and suggestions, and more minor points.

- The MS is scientifically accurate. See my comments about reporting of some of the statistical tests below
- Research methods adhere to the standards of the field. I had some points for clarification with respect to the interpretation of the results, below
- The paper is clearly written and logically structured and will be accessible to a broad scientific audience
- Figures and supplementary materials are appropriate; the supplementary materials provide the infant data as required by the journal
- The article is an appropriate length
- The empirical work adheres to the Declaration of Helsinki and ethical approval was given by the University of Oxford Medical Sciences Ethics Committee
- Data have been provided in the supplementary materials, and the model code is linked to in the author statements (this should be included in the proofs).
- Author contributions statement is transparent
- The data provided in the supplementary materials are sufficient to reproduce the analyses

Questions/suggestions:

1) I wasn't clear what the rationale for the dynamic learning rate was. We know that in this paradigm infants do increase their looking to novel stimuli, but that doesn't mean that they are learning more quickly - this is something we don't know. On page 8 it seems that the authors are arguing that the increase in learning rate is a proxy for looking time ("This is in line with the finding that infants' looking time during familiarisation is higher for more novel items, and this is interpreted as the time taken to incorporate that stimulus into the category representation [37, 39]."). If so, then how does this fit with the use of quantization error as a proxy for looking time on the test trials? Relatedly, how does this mechanism relate to findings from Kidd and colleagues that infants prefer to look at intermediate novelty stimuli (the "Goldilocks" studies, 2012, 2014)?

I'm not sure what order the work was done in (i.e., modelling first then empirical studies or vice versa). One possibility to make the logic of the paper clearer would be to present the empirical study first, then the model. This would allow the authors to argue that the critical, novel mechanism in the model that allows it to capture the empirical data is the adaptive learning rate; this would then have implications for theories of infant learning mechanisms. However I would not insist on this if the authors can clarify in some other way.

2) On page 9 the authors state: “Auto-encoders are trained by presenting all input patterns many times, and a core assumption is that the aim is to place equal weight on each stimulus, disregarding the sequence in which stimuli were presented (in other words, this approach is by definition unsuitable to model the particular memory effects we want to address).” In work from my own group (Twomey & Westermann, 2017; Dev. Sci.), we showed that an autoencoder could capture the results in M&P; so, autoencoders are also subject to sequence order effects. Whether they could capture the recency effects is a very interesting avenue for further research. The fact that two models have captured the same data doesn’t impact on the novelty of this paper, which I think makes a strong and interesting contribution. More broadly, in our paper we make some complementary arguments concerning the dynamic nature of categorisation that could strengthen the discussion of the current MS. In particular, we argue that plasticity of the network is key during learning; it seems to me this relates to the dynamic learning rate employed here. I’m usually very reluctant to ask authors to cite my own work but in this case I think it’s appropriate.

3) Although in the first set of simulations (Mather & Plunkett replication) the model shows greater error in the high ED condition, as per the stronger categorisation in that condition shown by infants in the original study, the preference scores in both conditions are less than 50%. Were they below chance? If so, how should this be interpreted? The results from the second set of simulations are quite different and above chance (although these t-tests aren’t reported either). Presumably this is due to the final stimulus manipulation; it would be great to see some discussion of this.

4) There are places where the logic of the paper feels circular. For example, on page 6, the authors set out the hypothesis that categorisation results are due to memory and priming based on inspection of the model and comparison with infant data, then at the bottom of page 7 state that they test this hypothesis in the model. Moving this discussion on page 7 to the GD could help. Then, on page 15, the authors state “To test the hypothesis that preferential looking at test is driven primarily by recency effects we set up a new set of simulations. The aim was (a) to understand the mathematical underpinnings of the recency effects” – but at this stage, there are no recency effects to simulate. Overall it wasn’t clear where they recency effects hypothesis comes from. Similar to my first question, it’s possible that this could be dealt with by restructuring the paper.

Minor points

- The MS switches between UK and US spelling
- P7: paragraphs beginning “We propose” and “Our model” could be combined
- P8: unnecessary exclamation mark in “(reducing the error!)”
- P16: in the footnote the authors state that no noise is introduced into the system at any point. However, the networks are randomly initialised, which does introduce noise – unless I’m misunderstanding, in which case apologies!
- Figure 5: there are no outliers in the model data so reference to this in the figure caption can be removed
- P24: There is no evidence for a preference for the peripheral stimulus on test trial 1 in the Dissimilar/Near condition ($p = .094$). This section should be rephrased to reflect this.
- P25: infants who saw the novel vs average test trial didn’t show a novelty preference ($ps = .075, .092$), so references to a trend should be removed. This is interesting in itself – later the authors argue that a null result at test in this paradigm could reflect a particular category structure, rather than a lack of learning. Is this what’s happening here?
- P27: re. the possibility of a primacy effect in the infant data, this could be investigated with the model, although this may be outside the scope of the current MS

Review form: Reviewer 2

Is the manuscript scientifically sound in its present form?

Yes

Are the interpretations and conclusions justified by the results?

No

Is the language acceptable?

Yes

Do you have any ethical concerns with this paper?

No

Have you any concerns about statistical analyses in this paper?

Yes

Recommendation?

Reject

Comments to the Author(s)

This paper present modeling and empirical data to investigate aspects of infant category learning. It argues that categorization is based on memory processes and thus, infants' responses in category learning tasks show recency effects.

The combination of modeling and experiment is commendable and offers the potential to reveal the mechanisms underlying category formation. The argument for taking memory processes into account is valuable as it can shed further light on the well-studied phenomenon of how infants learn object categories. The paper is not developmental per se (it has nothing to say about developmental processes or change) and the same arguments could be made for category learning in adults. But categorization in infants has been described as a fundamental process underlying many other cognitive abilities, so it is worthwhile to focus on infants here. The authors use a stimulus set that has been used in several earlier infant studies since the 1980s and this is a strength as there is data from different labs to show us how infants learn these items.

Yet, overall, idiosyncrasies in the model algorithm and relatively weak results for the infant studies make me less positive about the paper: in my view, the strong message that infants' looking at test in categorization tasks is driven by recency effects is not supported by either the model of the experimental data. I provide more detail on this and other issues below.

Introduction:

1. The conceptual novelty for which the authors argue (e.g., p 4 bottom) that "category activation does not merely involve invoking a preexisting abstract representation, but is inherently linked to memory" is in fact not very novel at all. Many papers have been written about the role of memory systems in category learning (see e.g., the work by Gregory Ashby), and this assumption is central to virtually all connectionist models (and there have been many on infant and adult categorization over the past 20 years, see e.g., the work by Mareschal and colleagues for infants and models by Rogers & McClelland for adults). There is also modeling work by Westermann & Mareschal (2012) that explicitly links infant category learning to the complementary memory system approach, with exemplar learning in one component and more global, long-term memory learning in another component. This work seems relevant here but is not mentioned.

Simulations:

3. The authors compare their SOM-based approach to the more widely used auto-encoder models of modeling infant category learning (p 9), but in doing so they misrepresent auto-encoder models. First, they state that auto-encoders are trained by presenting all items in a familiarization set many times with the core assumption that the aim is to place equal weight on each stimulus, and that these models therefore disregard the sequence in which stimuli are presented (the key concern in the present paper). However, most auto-encoder models tend to exactly replicate the familiarization phase in infants, presenting each item only once (e.g., French, Mermillod, Mareschal, & Quinn, 2004; Westermann & Mareschal, 2004). In fact there was a recent auto-encoder model (Twomey & Westermann, 2017) that modeled precisely the order effects found in Mather & Plunkett (2011) like in the present paper, and which the authors claim cannot be accounted for in such models.

Second, the authors argue that analyzing the internal representations of auto-encoders is more complicated than for SOMs and imply that such analyses in auto-encoders have little or no explanatory power (line 51). This claim is surprising as there are many papers that have used clustering of hidden representations and distance measures between the hidden representations for specific inputs, which is a straightforward process (e.g., McClelland & Rogers, 2003) that has often been used to explain the internal mechanisms of learning. It is not clear why the distance between multidimensional hidden representations in an auto-encoder should have less explanatory power than the distance between multidimensional BMU vectors on a feature map.

The comparison between SOMs and auto-encoders should therefore be more nuanced, and the results from the Twomey & Westermann paper, which modeled some of the same data as in the present paper, should be discussed and related to the present work.

4. The SOM algorithm used here only modifies the BMU and not a neighborhood of the BMU. This procedure will lead to the SOM to become an exemplar storage without a similarity structure across the map as is common in the typical algorithm when the neighborhood radius is gradually decreased across training. This doesn't therefore seem to be a very good category learning model since it is not able to represent internal category structure (i.e., similar category members will not be represented by neighboring map units).

5. Moreover, and importantly, this lack of neighborhood updating also make me suspect that, at least in the similar-transition sequence, but perhaps also in the dissimilar-transition sequence, there will be only one unit on the map that is the BMU for all eight familiarization stimuli. I believe this to be so because one unit will be the BMU for the first item and will have its weights adapted (rapidly, because of the variable learning rate), and all other units remain randomly initialized. It's therefore very likely that the same unit will be the BMU also for the second item, and so on (at least if the stimulus feature descriptions differ sufficiently from the random weight initializations). This single BMU will, at the end of familiarization, then be strongly adapted to the final familiarization item, leading to recency effects. In fact, the authors' explanation of the recency effects (p 17) completely relies on a BMU being shared between most or all exemplars: if a different BMU responded to each exemplar, then the order in which they were presented would not matter (as, in the absence of a neighborhood function, only the BMU is updated without any effect on other units). Although the authors suggest (p 18 top) that for the peripheral test stimulus a different BMU to the last training item will be selected, this is not necessary: even if there is a single BMU shared by all training and test items, the quantization error for a peripheral item would still be higher after 'near' final training item. I expect that the model would produce exactly the same results if it only had a single unit.

Importantly, these are not merely technical details: the implication of this explanation is that the model cannot *discriminate* between the familiarization items (since they all have the same BMU), whereas the process of categorization is based on the idea that objects can be discriminated but are nevertheless treated as equivalent, and infants have often been shown that they can discriminate between items for which they form a category. – To clarify this, at the very

least the authors need to report across how many BMUs the familiarization items were distributed in both training conditions and what this means for categorization.

Infant experiments:

6. A problem is that infants familiarized to the training stimuli only in one of the four conditions (sim/near). And in fact this is the only group showing a looking preference at test (there is a weak trend - $p = .094$ - for the Diss/Near condition).

7. Interestingly the second test trial is the opposite of the first one, with the only effect now a preference for the peripheral item in the Diss/Far condition. The authors rightly note that this second test trial was not modeled (although it would be interesting and straightforward to do so), but it would be good to at least speculate how this result fits in with the narrative of recency effects - from my understanding it directly contradicts this narrative. The authors do discuss this result (p 26f) but their explanation critically assumes that infants have formed a category, but the fact that they did not familiarize to the training stimuli makes this assumption questionable.

8. Overall, the results from infants seem to lend only weak support for recency effects, which occurred in only one of the four conditions (and a weak trend in a second), and only on the first test trial. If anything, the data only hint at presentation order having an effect on looking at test (which was already shown by Mather & Plunkett, 2011) with the role of the final item having only a weak, if any, additional effect.

9. Together, these results are much less strong than the narrative makes us believe (that "perceived novelty depends mainly on similarity to the immediately preceding item" (abstract), "that the recency effect determines looking" (p 26) and that "recency effects are dominant in infant categorization" (p 27) - if anything, order effects interact with a more global learning of category structure; and that "infants exhibited a clear preference" (p 26) - they only did so in one of the "far" conditions and in no other, and only on the first test trial). The data should be discussed much more carefully and the narrative toned down.

The conclusions that familiarization sequences should be taken seriously is an important one, as shown in Mather & Plunkett (2011), but it is typically addressed by familiarizing different infants with different sequences.

9. The issue of p values: the looking preference result for the Diss/Near condition comes out with $p = .094$ (p 25) and is interpreted as a "clear preference" (p 26); the result for looking time collapsed across conditions (p 27) comes out with $p = .097$ and is described as not differing from chance. It would be better to adopt a consistent approach to describing results as significant or not (ideally, the conventional 0.05). Of course frequentist statistics have their flaws, and perhaps the results should be supplemented by Bayesian analyses that also allow for interpretation of null results.

10. The collapsed results reported on p 26 show no effect for both the similar and dissimilar-transition conditions, although to my understanding this would be a direct replication of Mather & Plunkett (2011). Is this non-replication of the earlier results a concern?

Minor points:

P 8 line 25: the link between novelty/familiarity/no preference and 'model quantisation error' should probably be explained here (or should be rephrased).

References:

McClelland, J. L., & Rogers, T. T. (2003). The parallel distributed processing approach to semantic cognition. *Nature Reviews Neuroscience*, 4(4), 310-322.

French, R. M., Mareschal, D., Mermillod, M., Quinn, P.C. (2004). The role of bottom-up processing in perceptual categorization by 3-to 4-month-old infants: Simulations and data. *Journal of Experimental Psychology: General*, 133(3), 382–397.

Twomey, K. E., & Westermann, G. (2017). Curiosity-based learning in infants: a neurocomputational approach. *Developmental Science*, 5(2), e12629–13.

Westermann, G., & Mareschal, D. (2012). Mechanisms of developmental change in infant categorization. *Cognitive Development*, 27(4), 367–382.

Decision letter (RSOS-200328.R0)

Dear Dr Althaus,

The editors assigned to your paper ("Infants' categorisation as a dynamic process inherently linked to memory") have now received comments from reviewers. We would like you to revise your paper in accordance with the referee and Associate Editor suggestions which can be found below (not including confidential reports to the Editor). Please note this decision does not guarantee eventual acceptance.

Please submit a copy of your revised paper before 15-May-2020. Please note that the revision deadline will expire at 00.00am on this date. If we do not hear from you within this time then it will be assumed that the paper has been withdrawn. In exceptional circumstances, extensions may be possible if agreed with the Editorial Office in advance. We do not allow multiple rounds of revision so we urge you to make every effort to fully address all of the comments at this stage. If deemed necessary by the Editors, your manuscript will be sent back to one or more of the original reviewers for assessment. If the original reviewers are not available, we may invite new reviewers.

If your study uses humans or animals please include details of the ethical approval received, including the name of the committee that granted approval. For human studies please also detail

whether informed consent was obtained. For field studies on animals please include details of all permissions, licences and/or approvals granted to carry out the fieldwork.

- Data accessibility

If you wish to submit your supporting data or code to Dryad (<http://datadryad.org/>), or modify your current submission to dryad, please use the following link:
<http://datadryad.org/submit?journalID=RSOS&manu=RSOS-200328>

- Competing interests

- Authors' contributions

- Acknowledgements

- Funding statement

on behalf of Dr Teodora Gliga (Associate Editor)
 openscience@royalsociety.org

Associate Editor's comments (Dr Teodora Gliga):

Associate Editor: 1

Comments to the Author:

I have now received comments from two experts in the field. Both agree that you address an important question but vary in whether they consider your work novel. When resubmitting your manuscript, please make sure to give a point by point response to each reviewer's comment. Please pay particular attention when addressing the following issues:

1. Both reviewers commented on your potential misinterpretation of the scope of auto-encoder models and made particular reference to a paper that may have also modelled order effects (Twomey & Westermann, 2017). If you agree with their comments, please further clarify how your model differs.
2. There is lack of clarity in terms of whether you think looking time during familiarisation, in particular a decrease from block 1 to 2, reflects learning, in your paradigm. A distinction exists in the infancy literature between familiarisation studies and habituation studies, the latter presenting more stimuli until a decrease in looking (to a criterion) is observed. Given you use a familiarisation design, your interpretation of the change in looking between blocks does not seem justified.
3. I think that it is justified to first present the model and then the study inspired by this model's results. However, there is some confusion in terms of how much work had been done already since you refer to preliminary observations made on M & P dataset. Please clarify how the current work is different from that preliminary/exploratory analysis.
4. Finally, I concur with reviewer 2 that formal modelling of effects of test 1 on test 2 would be illuminating but I also expect this to be a difficult endeavour, given individual differences in looking during test 1. At the very least, please comment on these challenges and on how behaviour in test 1 may explain performance in test 2.

Comments to Author:

Reviewers' Comments to Author:

Reviewer: 1

Comments to the Author(s)

The manuscript describes a computational model and an empirical study investigating categorisation in infancy. Using a SOM-based model, the authors first capture data from an existing empirical study (Mather & Plunkett, 2010), which demonstrated that the order in which 10-month-old infants encounter category exemplars during familiarisation affects their performance at test. Next, the authors used the model to demonstrate that, rather than forming a category representation in which the influence of all exemplars is equal, due to the model's adaptive learning rate the representation is most heavily influenced by the final familiarisation exemplar, resulting in recency effects in test performance. The authors capture this prediction in 10-month-old infants, concluding that infant categorisation is a dynamic, online process, and discussing implications for study design and interpretation.

Overall the paper deals with an important topic of current interest in the field. It will make a strong contribution in terms of theories of infants' online learning and offers a novel computational mechanism for modelling these processes. The model predictions were tested empirically, which is the gold standard in the computational modelling subfield of psychology, and taken together the model and empirical data have both theoretical and methodological implications. Below I have responded to the specific points in the reviewer guidelines for this journal, followed by some questions and suggestions, and more minor points.

- The MS is scientifically accurate. See my comments about reporting of some of the statistical tests below
- Research methods adhere to the standards of the field. I had some points for clarification with respect to the interpretation of the results, below
- The paper is clearly written and logically structured and will be accessible to a broad scientific audience
- Figures and supplementary materials are appropriate; the supplementary materials provide the infant data as required by the journal
- The article is an appropriate length
- The empirical work adheres to the Declaration of Helsinki and ethical approval was given by the University of Oxford Medical Sciences Ethics Committee
- Data have been provided in the supplementary materials, and the model code is linked to in the author statements (this should be included in the proofs).
- Author contributions statement is transparent
- The data provided in the supplementary materials are sufficient to reproduce the analyses

Questions/suggestions:

1) I wasn't clear what the rationale for the dynamic learning rate was. We know that in this paradigm infants do increase their looking to novel stimuli, but that doesn't mean that they are learning more quickly – this is something we don't know. On page 8 it seems that the authors are arguing that the increase in learning rate is a proxy for looking time (“This is in line with the finding that infants’ looking time during familiarisation is higher for more novel items, and this is interpreted as the time taken to incorporate that stimulus into the category representation [37, 39].”). If so, then how does this fit with the use of quantization error as a proxy for looking time on the test trials? Relatedly, how does this mechanism relate to findings from Kidd and colleagues that infants prefer to look at intermediate novelty stimuli (the “Goldilocks” studies, 2012, 2014)?

I'm not sure what order the work was done in (i.e., modelling first then empirical studies or vice versa). One possibility to make the logic of the paper clearer would be to present the empirical study first, then the model. This would allow the authors to argue that the critical, novel mechanism in the model that allows it to capture the empirical data is the adaptive learning rate; this would then have implications for theories of infant learning mechanisms. However I would not insist on this if the authors can clarify in some other way.

2) On page 9 the authors state: “Auto-encoders are trained by presenting all input patterns many times, and a core assumption is that the aim is to place equal weight on each stimulus, disregarding the sequence in which stimuli were presented (in other words, this approach is by definition unsuitable to model the particular memory effects we want to address).” In work from my own group (Twomey & Westermann, 2017; Dev. Sci.), we showed that an autoencoder could capture the results in M&P; so, autoencoders are also subject to sequence order effects. Whether they could capture the recency effects is a very interesting avenue for further research. The fact that two models have captured the same data doesn't impact on the novelty of this paper, which I think makes a strong and interesting contribution. More broadly, in our paper we make some complementary arguments concerning the dynamic nature of categorisation that could strengthen the discussion of the current MS. In particular, we argue that plasticity of the network is key during learning; it seems to me this relates to the dynamic learning rate employed here. I'm usually very reluctant to ask authors to cite my own work but in this case I think it's appropriate.

3) Although in the first set of simulations (Mather & Plunkett replication) the model shows greater error in the high ED condition, as per the stronger categorisation in that condition shown by infants in the original study, the preference scores in both conditions are less than 50%. Were they below chance? If so, how should this be interpreted? The results from the second set of

simulations are quite different and above chance (although these t-tests aren't reported either). Presumably this is due to the final stimulus manipulation; it would be great to see some discussion of this.

4) There are places where the logic of the paper feels circular. For example, on page 6, the authors set out the hypothesis that categorisation results are due to memory and priming based on inspection of the model and comparison with infant data, then at the bottom of page 7 state that they test this hypothesis in the model. Moving this discussion on page 7 to the GD could help. Then, on page 15, the authors state "To test the hypothesis that preferential looking at test is driven primarily by recency effects we set up a new set of simulations. The aim was (a) to understand the mathematical underpinnings of the recency effects" - but at this stage, there are no recency effects to simulate. Overall it wasn't clear where they recency effects hypothesis comes from. Similar to my first question, it's possible that this could be dealt with by restructuring the paper.

Minor points

- The MS switches between UK and US spelling
- P7: paragraphs beginning "We propose" and "Our model" could be combined
- P8: unnecessary exclamation mark in "(reducing the error!)"
- P16: in the footnote the authors state that no noise is introduced into the system at any point. However, the networks are randomly initialised, which does introduce noise - unless I'm misunderstanding, in which case apologies!
- Figure 5: there are no outliers in the model data so reference to this in the figure caption can be removed
- P24: There is no evidence for a preference for the peripheral stimulus on test trial 1 in the Dissimilar/Near condition ($p = .094$). This section should be rephrased to reflect this.
- P25: infants who saw the novel vs average test trial didn't show a novelty preference ($p_s = .075, .092$), so references to a trend should be removed. This is interesting in itself - later the authors argue that a null result at test in this paradigm could reflect a particular category structure, rather than a lack of learning. Is this what's happening here?
- P27: re. the possibility of a primacy effect in the infant data, this could be investigated with the model, although this may be outside the scope of the current MS

Reviewer: 2

Comments to the Author(s)

This paper present modeling and empirical data to investigate aspects of infant category learning. It argues that categorization is based on memory processes and thus, infants' responses in category learning tasks show recency effects.

The combination of modeling and experiment is commendable and offers the potential to reveal the mechanisms underlying category formation. The argument for taking memory processes into account is valuable as it can shed further light on the well-studied phenomenon of how infants learn object categories. The paper is not developmental per se (it has nothing to say about developmental processes or change) and the same arguments could be made for category learning in adults. But categorization in infants has been described as a fundamental process underlying many other cognitive abilities, so it is worthwhile to focus on infants here. The authors use a stimulus set that has been used in several earlier infant studies since the 1980s and this is a strength as there is data from different labs to show us how infants learn these items.

Yet, overall, idiosyncrasies in the model algorithm and relatively weak results for the infant studies make me less positive about the paper: in my view, the strong message that infants' looking at test in categorization tasks is driven by recency effects is not supported by either the model or the experimental data. I provide more detail on this and other issues below.

Introduction:

1. The conceptual novelty for which the authors argue (e.g., p 4 bottom) that “category activation does not merely involve invoking a preexisting abstract representation, but is inherently linked to memory” is in fact not very novel at all. Many papers have been written about the role of memory systems in category learning (see e.g., the work by Gregory Ashby), and this assumption is central to virtually all connectionist models (and there have been many on infant and adult categorization over the past 20 years, see e.g., the work by Mareschal and colleagues for infants and models by Rogers & McClelland for adults). There is also modeling work by Westermann & Mareschal (2012) that explicitly links infant category learning to the complementary memory system approach, with exemplar learning in one component and more global, long-term memory learning in another component. This work seems relevant here but is not mentioned.

Simulations:

3. The authors compare their SOM-based approach to the more widely used auto-encoder models of modeling infant category learning (p 9), but in doing so they misrepresent auto-encoder models. First, they state that auto-encoders are trained by presenting all items in a familiarization set many times with the core assumption that the aim is to place equal weight on each stimulus, and that these models therefore disregard the sequence in which stimuli are presented (the key concern in the present paper). However, most auto-encoder models tend to exactly replicate the familiarization phase in infants, presenting each item only once (e.g., French, Mermillod, Mareschal, & Quinn, 2004; Westermann & Mareschal, 2004). In fact there was a recent auto-encoder model (Twomey & Westermann, 2017) that modeled precisely the order effects found in Mather & Plunkett (2011) like in the present paper, and which the authors claim cannot be accounted for in such models.

Second, the authors argue that analyzing the internal representations of auto-encoders is more complicated than for SOMs and imply that such analyses in auto-encoders have little or no explanatory power (line 51). This claim is surprising as there are many papers that have used clustering of hidden representations and distance measures between the hidden representations for specific inputs, which is a straightforward process (e.g., McClelland & Rogers, 2003) that has often been used to explain the internal mechanisms of learning. It is not clear why the distance between multidimensional hidden representations in an auto-encoder should have less explanatory power than the distance between multidimensional BMU vectors on a feature map.

The comparison between SOMs and auto-encoders should therefore be more nuanced, and the results from the Twomey & Westermann paper, which modeled some of the same data as in the present paper, should be discussed and related to the present work.

4. The SOM algorithm used here only modifies the BMU and not a neighborhood of the BMU. This procedure will lead to the SOM to become an exemplar storage without a similarity structure across the map as is common in the typical algorithm when the neighborhood radius is gradually decreased across training. This doesn't therefore seem to be a very good category learning model since it is not able to represent internal category structure (i.e., similar category members will not be represented by neighboring map units).

5. Moreover, and importantly, this lack of neighborhood updating also make me suspect that, at least in the similar-transition sequence, but perhaps also in the dissimilar-transition sequence, there will be only one unit on the map that is the BMU for all eight familiarization stimuli. I believe this to be so because one unit will be the BMU for the first item and will have its weights adapted (rapidly, because of the variable learning rate), and all other units remain randomly initialized. It's therefore very likely that the same unit will be the BMU also for the second item, and so on (at least if the stimulus feature descriptions differ sufficiently from the random weight initializations). This single BMU will, at the end of familiarization, then be strongly adapted to

the final familiarization item, leading to recency effects. In fact, the authors' explanation of the recency effects (p 17) completely relies on a BMU being shared between most or all exemplars: if a different BMU responded to each exemplar, then the order in which they were presented would not matter (as, in the absence of a neighborhood function, only the BMU is updated without any effect on other units). Although the authors suggest (p 18 top) that for the peripheral test stimulus a different BMU to the last training item will be selected, this is not necessary: even if there is a single BMU shared by all training and test items, the quantization error for a peripheral item would still be higher after 'near' final training item. I expect that the model would produce exactly the same results if it only had a single unit.

Importantly, these are not merely technical details: the implication of this explanation is that the model cannot *discriminate* between the familiarization items (since they all have the same BMU), whereas the process of categorization is based on the idea that objects can be discriminated but are nevertheless treated as equivalent, and infants have often been shown that they can discriminate between items for which they form a category. – To clarify this, at the very least the authors need to report across how many BMUs the familiarization items were distributed in both training conditions and what this means for categorization.

Infant experiments:

6. A problem is that infants familiarized to the training stimuli only in one of the four conditions (sim/near). And in fact this is the only group showing a looking preference at test (there is a weak trend – $p = .094$ – for the Diss/Near condition).

7. Interestingly the second test trial is the opposite of the first one, with the only effect now a preference for the peripheral item in the Diss/Far condition. The authors rightly note that this second test trial was not modeled (although it would be interesting and straightforward to do so), but it would be good to at least speculate how this result fits in with the narrative of recency effects – from my understanding it directly contradicts this narrative. The authors do discuss this result (p 26f) but their explanation critically assumes that infants have formed a category, but the fact that they did not familiarize to the training stimuli makes this assumption questionable.

8. Overall, the results from infants seem to lend only weak support for recency effects, which occurred in only one of the four conditions (and a weak trend in a second), and only on the first test trial. If anything, the data only hint at presentation order having an effect on looking at test (which was already shown by Mather & Plunkett, 2011) with the role of the final item having only a weak, if any, additional effect.

9. Together, these results are much less strong than the narrative makes us believe (that “perceived novelty depends mainly on similarity to the immediately preceding item” (abstract), “that the recency effect determines looking” (p 26) and that “recency effects are dominant in infant categorization” (p 27) – if anything, order effects interact with a more global learning of category structure; and that “infants exhibited a clear preference” (p 26) – they only did so in one of the “far” conditions and in no other, and only on the first test trial). The data should be discussed much more carefully and the narrative toned down.

The conclusions that familiarization sequences should be taken seriously is an important one, as shown in Mather & Plunkett (2011), but it is typically addressed by familiarizing different infants with different sequences.

9. The issue of p values: the looking preference result for the Diss/Near condition comes out with $p = .094$ (p 25) and is interpreted as a “clear preference” (p 26); the result for looking time collapsed across conditions (p 27) comes out with $p = .097$ and is described as not differing from chance. It would be better to adopt a consistent approach to describing results as significant or not (ideally, the conventional 0.05). Of course frequentist statistics have their flaws, and perhaps

the results should be supplemented by Bayesian analyses that also allow for interpretation of null results.

10. The collapsed results reported on p 26 show no effect for both the similar and dissimilar-transition conditions, although to my understanding this would be a direct replication of Mather & Plunkett (2011). Is this non-replication of the earlier results a concern?

Minor points:

P 8 line 25: the link between novelty/familiarity/no preference and 'model quantisation error' should probably be explained here (or should be rephrased).

References:

McClelland, J. L., & Rogers, T. T. (2003). The parallel distributed processing approach to semantic cognition. *Nature Reviews Neuroscience*, 4(4), 310–322.

French, R. M., Mareschal, D., Mermillod, M., Quinn, P.C. (2004). The role of bottom-up processing in perceptual categorization by 3-to 4-month-old infants: Simulations and data. *Journal of Experimental Psychology: General*, 133(3), 382–397.

Twomey, K. E., & Westermann, G. (2017). Curiosity-based learning in infants: a neurocomputational approach. *Developmental Science*, 5(2), e12629–13.

Westermann, G., & Mareschal, D. (2012). Mechanisms of developmental change in infant categorization. *Cognitive Development*, 27(4), 367–382.

Author's Response to Decision Letter for (RSOS-200328.R0)

See Appendix A.

RSOS-200328.R1 (Revision)

Review form: Reviewer 1

Is the manuscript scientifically sound in its present form?

Yes

Are the interpretations and conclusions justified by the results?

Yes

Is the language acceptable?

Yes

Do you have any ethical concerns with this paper?

No

Have you any concerns about statistical analyses in this paper?

No

Recommendation?

Accept as is

Comments to the Author(s)

This is a review of a manuscript I have reviewed once previously. I apologise to the authors and editor for the delay in getting the review back to them.

The authors have addressed my comments thoughtfully and thoroughly. The rationale for and chronology of the studies are now much clearer, and the authors now provide a more nuanced discussion of autoencoder vs. SOM approaches. The conclusions the authors draw are now more conservative, and my major questions have been addressed.

On this basis I have no further questions. I would however ask that the authors share their model and analysis code, as per the journal's requirements.

Review form: Reviewer 2**Is the manuscript scientifically sound in its present form?**

Yes

Are the interpretations and conclusions justified by the results?

Yes

Is the language acceptable?

Yes

Do you have any ethical concerns with this paper?

No

Have you any concerns about statistical analyses in this paper?

No

Recommendation?

Major revision is needed (please make suggestions in comments)

Comments to the Author(s)

The paper is now much improved and the authors have done an excellent job of addressing many of the raised points in detail.

Most of my remaining points concern improving the transparency of the model. I do believe that the authors need to be more explicit about the contribution made by the model to explaining the recency effect. Relatedly, the comparison between the present model and more typical SOMs lacks detail. Specifically:

1. The comparison between better known standard SOMs and the current model is lacking in detail. Clearly, the lack of a neighborhood radius in the model is a major departure from standard SOMs because without it, no topologies are formed, and these are a core property of standard SOMs. In the discussion of the differences between the current model and a typical SOM (p10), this should be mentioned. Again on p 15, the authors state that the main difference between theirs and standard SOMs is the adaptive learning rate, and the lack of a neighborhood radius is not explicitly mentioned but just described by the Kronecker function (p. 15) – this important detail will escape most readers not familiar with SOMs. It is therefore important that the lack of a

neighborhood update scheme is spelled out clearly as a difference to typical SOMs, as it greatly affects the functioning of the model.

2. The explanation of how the model learns categories on p 10 relies on a non-zero neighborhood function ('.. because that unit has been moved repeatedly towards familiarization exemplars' and '...because no unit has been moved in that direction'). While this explanation is correct for typical SOMs with neighborhood updating, the explanation in the zero-neighborhood model (which does not develop topological representations) is rather different, namely, that a similar item shares the same BMU with training exemplars, and a dissimilar item might not. It should be made clearer in the paper that typical SOMs and the current model learn categories in these different ways.

3. In the response letter the authors state that not all of the models had only one activated BMU - does this mean that most did? It should be reported over how many units the BMUs are distributed on average.

As I wrote in my review of the previous version of the paper, the recency effects (and order effects in general) in the model in my view crucially depend on the sharing of the BMU by most or all training and test exemplars, since there is no neighborhood updating. In my view this means that recency effects arise because the model does not discriminate between different category exemplars - if two stimuli share a BMU then the model does not discriminate between them. The authors might disagree, but this should be discussed. To my reading this prediction - that recency effect arise because the model does not discriminate between specific stimuli - is the main contribution of the model as it is a clear prediction that can be tested.

4. On p 19f the authors explain intuitively how recency effects arise in the model. They state that for a far test stimulus after a near final training item a different BMU will be selected - was this the case in the model? It would improve the transparency of the model if this was analyzed.

5. In Simulation 1 the order effects from Mather & Plunkett (2011) are replicated, but the authors also state that the final familiarization stimulus had an effect. Were these effects independent, or were they driven by recency effects? That is, were the dissimilar-transition sequences more likely to end in a 'near' item than the dissimilar-transition sequences? This would be important to know to understand whether the model formed more global sequence representations or if it was entirely driven by local transitions between the final training item and the test item. Perhaps the authors can add some detail about this at the end of that section.

6. It would be useful to report (and interpret) the Bayes factors that were calculated in the response letter also in the paper.

Decision letter (RSOS-200328.R1)

Dear Dr Althaus,

On behalf of the Editors, we are pleased to inform you that your Manuscript RSOS-200328.R1 "Infants' categorisation as a dynamic process inherently linked to memory" has been accepted for publication in Royal Society Open Science subject to minor revision in accordance with the referees' reports. Please find the referees' comments along with any feedback from the Editors below my signature.

Please submit your revised manuscript and required files (see below) no later than 7 days from today's (ie 08-Sep-2020) date. Note: the ScholarOne system will 'lock' if submission of the revision is attempted 7 or more days after the deadline. If you do not think you will be able to meet this deadline please contact the editorial office immediately.

on behalf of Dr Teodora Gliga (Associate Editor)
openscience@royalsociety.org

Associate Editor Comments to Author (Dr Teodora Gliga):

Your paper has now been seen by the two reviewers that provided initial feed-back - they both find your manuscript much improved, with clearer justifications and methodology and I concur with their views. I particularly liked how by restructuring the manuscript you convey the dynamic interplay between computational and empirical work. Reviewer 2 still requires some additional clarifications which you will have to provide when re-submitting your manuscript.

Reviewer comments to Author:

Reviewer: 1
Comments to the Author(s)

This is a review of a manuscript I have reviewed once previously. I apologise to the authors and editor for the delay in getting the review back to them.

The authors have addressed my comments thoughtfully and thoroughly. The rationale for and chronology of the studies are now much clearer, and the authors now provide a more nuanced discussion of autoencoder vs. SOM approaches. The conclusions the authors draw are now more conservative, and my major questions have been addressed.

On this basis I have no further questions. I would however ask that the authors share their model and analysis code, as per the journal's requirements.

Reviewer: 2

Comments to the Author(s)

The paper is now much improved and the authors have done an excellent job of addressing many of the raised points in detail.

Most of my remaining points concern improving the transparency of the model. I do believe that the authors need to be more explicit about the contribution made by the model to explaining the recency effect. Relatedly, the comparison between the present model and more typical SOMs lacks detail. Specifically:

1. The comparison between better known standard SOMs and the current model is lacking in detail. Clearly, the lack of a neighborhood radius in the model is a major departure from standard SOMs because without it, no topologies are formed, and these are a core property of standard SOMs. In the discussion of the differences between the current model and a typical SOM (p10), this should be mentioned. Again on p 15, the authors state that the main difference between theirs and standard SOMs is the adaptive learning rate, and the lack of a neighborhood radius is not explicitly mentioned but just described by the Kronecker function (p. 15) – this important detail will escape most readers not familiar with SOMs. It is therefore important that the lack of a neighborhood update scheme is spelled out clearly as a difference to typical SOMs, as it greatly affects the functioning of the model.

2. The explanation of how the model learns categories on p 10 relies on a non-zero neighborhood function ('.. because that unit has been moved repeatedly towards familiarization exemplars' and '...because no unit has been moved in that direction'). While this explanation is correct for typical SOMs with neighborhood updating, the explanation in the zero-neighborhood model (which does not develop topological representations) is rather different, namely, that a similar item shares the same BMU with training exemplars, and a dissimilar item might not. It should be made clearer in the paper that typical SOMs and the current model learn categories in these different ways.

3. In the response letter the authors state that not all of the models had only one activated BMU – does this mean that most did? It should be reported over how many units the BMUs are distributed on average.

As I wrote in my review of the previous version of the paper, the recency effects (and order effects in general) in the model in my view crucially depend on the sharing of the BMU by most or all training and test exemplars, since there is no neighborhood updating. In my view this means that recency effects arise because the model does not discriminate between different category exemplars – if two stimuli share a BMU then the model does not discriminate between them. The authors might disagree, but this should be discussed. To my reading this prediction – that recency effect arise because the model does not discriminate between specific stimuli - is the main contribution of the model as it is a clear prediction that can be tested.

4. On p 19f the authors explain intuitively how recency effects arise in the model. They state that for a far test stimulus after a near final training item a different BMU will be selected – was this the case in the model? It would improve the transparency of the model if this was analyzed.

5. In Simulation 1 the order effects from Mather & Plunkett (2011) are replicated, but the authors also state that the final familiarization stimulus had an effect. Were these effects independent, or were they driven by recency effects? That is, were the dissimilar-transition sequences more likely to end in a 'near' item than the dissimilar-transition sequences? This would be important to know to understand whether the model formed more global sequence representations or if it was entirely driven by local transitions between the final training item and the test item. Perhaps the authors can add some detail about this at the end of that section.

6. It would be useful to report (and interpret) the Bayes factors that were calculated in the response letter also in the paper.

===PREPARING YOUR MANUSCRIPT===

- one version identifying all the changes that have been made (for instance, in coloured highlight, in bold text, or tracked changes);
- a 'clean' version of the new manuscript that incorporates the changes made, but does not highlight them.

 This version will be used for typesetting.

===PREPARING YOUR REVISION IN SCHOLARONE===

- An individual file of each figure (EPS or print-quality PDF preferred [either format should be produced directly from original creation package], or original software format).
 - An editable file of each table (.doc, .docx, .xls, .xlsx, or .csv).
 - An editable file of all figure and table captions.
- Note: you may upload the figure, table, and caption files in a single Zip folder.
- Any electronic supplementary material (ESM).
 - If you are requesting a discretionary waiver for the article processing charge, the waiver form must be included at this step.
 - If you are providing image files for potential cover images, please upload these at this step, and inform the editorial office you have done so. You must hold the copyright to any image provided.
 - A copy of your point-by-point response to referees and Editors. This will expedite the preparation of your proof.

- Ensure that your data access statement meets the requirements at <https://royalsociety.org/journals/authors/author-guidelines/#data>. You should ensure that you cite the dataset in your reference list. If you have deposited data etc in the Dryad repository, please only include the 'For publication' link at this stage. You should remove the 'For review' link.
- If you are requesting an article processing charge waiver, you must select the relevant waiver option (if requesting a discretionary waiver, the form should have been uploaded at Step 3 'File upload' above).
- If you have uploaded ESM files, please ensure you follow the guidance at <https://royalsociety.org/journals/authors/author-guidelines/#supplementary-material> to include a suitable title and informative caption. An example of appropriate titling and captioning may be found at https://figshare.com/articles/Table_S2_from_Is_there_a_trade-off_between_peak_performance_and_performance_breadth_across_temperatures_for_aerobic_scope_in_teleost_fishes_/3843624.

Author's Response to Decision Letter for (RSOS-200328.R1)

See Appendix B.

Decision letter (RSOS-200328.R2)

Dear Dr Althaus,

It is a pleasure to accept your manuscript entitled "Infant categorisation as a dynamic process linked to memory" in its current form for publication in Royal Society Open Science.

on behalf of Dr Teodora Gliga (Associate Editor)
openscience@royalsociety.org

Appendix A

Royal Society Open Science Manuscript ID RSOS-200328
Responses to editor & reviewers

I. Replies to the Associate Editor:

1. *Both reviewers commented on your potential misinterpretation of the scope of auto-encoder models and made particular reference to a paper that may have also modelled order effects (Twomey & Westermann, 2017). If you agree with their comments, please further clarify how your model differs.*

Reply I.1: We agree with the reviewers' comments (see Replies II.3 and III.2 below), and have revised the section on p. 11 accordingly. This includes Twomey & Westermann's (2017)'s work. Ultimately we believe SOMs and autoencoders are equally valid approaches that converge in terms of their findings with regard to simulating infant familiarisation studies, as our preliminary model and Twomey & Westermann's (2017) results demonstrate.

We chose to use SOMs because of their biological and psychological plausibility, and because they have already proven to be useful tools to explain infant categorisation phenomena (as for instance in Gliozzi, Mayor, Hu & Plunkett, 2009).

2. *There is lack of clarity in terms of whether you think looking time during familiarisation, in particular a decrease from block 1 to 2, reflects learning, in your paradigm. A distinction exists in the infancy literature between familiarisation studies and habituation studies, the latter presenting more stimuli until a decrease in looking (to a criterion) is observed. Given you use a familiarisation design, your interpretation of the change in looking between blocks does not seem justified.*

Reply I.2:

We are of course aware of the distinction between familiarisation and habituation paradigms and agree that this has to be clear in the manuscript. Familiarisation techniques simply allow the infant to examine the stimuli, rather than requiring a decrease in looking time. However, the familiarisation/novelty preference task typically depends on behaviour at test, not during familiarisation. In many such experiments there is no decrease in looking time across familiarisation but a distinct novelty preference is reported. If there is no other explanation of why infants should prefer one test stimulus over the other, this can still be interpreted as evidence for learning (e.g. Quinn et al., 1993). In our manuscript we conclude learning *has occurred* in those conditions for which we find a preference for the peripheral item.

Note, however, our discussion of the *plausibility* of learning occurring even in the remaining conditions (page 27/28), on the basis that the model in fact predicts decreased preferential looking in the "Far" conditions despite successful learning of a category representation (due to the category representation being shifted to

a location between average and peripheral item). We have emphasised here that this is a discussion on the basis of model results rather than experimental evidence (p. 28, line 3 ff.).

We have made adjustments in the manuscript to remove any reference to “habituate” or “habituation” in the context of our study (on p. 23-24 we changed “the degree of infant habituation” to “the degree of infant familiarisation” and “habituated to a greater degree” to “familiarised to a greater degree”). We also clarified the difference between familiarisation and habituation paradigms on pages 5/6.

3. *I think that it is justified to first present the model and then the study inspired by this model's results. However, there is some confusion in terms of how much work had been done already since you refer to preliminary observations made on M & P dataset. Please clarify how the current work is different from that preliminary/exploratory analysis.*

Reply I.3:

We have changed the structure to make the difference between our preliminary experiment and the main experiment clear – see Reply RII.2 below.

We began by simulating Mather & Plunkett’s (2011) results, using an existing model introduced by GIoZZi, Mayor, Hu, & Plunkett (2009). This is the first part of our exploration, now described as Simulation 1.

These simulations led to the observation of recency effects, which could explain Mather & Plunkett’s results (this was not addressed in their original paper).

In order to explore this systematically and generate new predictions that would be testable with infants, we then constructed a new set of sequences in a 2x2 design, taking into consideration both the role of euclidean distance and recency effects.

Simulation 2 (using this new 2x2 design) was run prior to our infant experiments, i.e. was used to generate predictions. The infant experiment was conducted last (described in section “Infant Experiment”). Its results confirm the model’s ability to accurately predict infant behaviour.

We have included one section (“Rationale: Hypothesizing recency effects”) prior to the simulation and experimental section in order to explain this timeline of our reasoning and step-wise approach, and also provided some more insight into the earlier processes leading up to this work in the section leading up to this, on p. 7/8. We hope that this is now clear in the manuscript.

4. *Finally, I concur with reviewer 2 that formal modelling of effects of test 1 on test 2 would be illuminating but I also expect this to be a difficult endeavour, given individual differences in looking during test 1. At the very least, please comment on these challenges and on how behaviour in test 1 may explain performance in test 2.*

Reply I.4: As we argue in Reply III.7A) to reviewer 2, it is in fact not straight

forward to model the second test trial, because infants are exposed to two items simultaneously during the first trial. Any attempt of incorporating this would involve making assumptions in the model about what simultaneous exposure implies for infants. Since infants may switch back and forth between items on test trial 1, this would need to be modelled by a flexible sequence with flexible exposure duration. However, there is also experimental evidence that there is a learning advantage in a trial that allows the direct comparison of two items, but how to implement this is nontrivial. We agree that there could be interesting insights gained from a modelling process in this direction, but this would be a new modelling project that is rather beyond the scope of the present paper.

II. Replies to Reviewer 1:

1. *I wasn't clear what the rationale for the dynamic learning rate was. We know that in this paradigm infants do increase their looking to novel stimuli, but that doesn't mean that they are learning more quickly – this is something we don't know. On page 8 it seems that the authors are arguing that the increase in learning rate is a proxy for looking time (“This is in line with the finding that infants' looking time during familiarisation is higher for more novel items, and this is interpreted as the time taken to incorporate that stimulus into the category representation [37, 39].”). If so, then how does this fit with the use of quantization error as a proxy for looking time on the test trials? Relatedly, how does this mechanism relate to findings from Kidd and colleagues that infants prefer to look at intermediate novelty stimuli (the “Goldilocks” studies, 2012, 2014)?*

Reply II.1: The model inherits its properties, as the dynamic learning rate, from its original version proposed by Gliozzi et al. (2009), where the rationale was that the learning rate depends on attention, which itself depends on the novelty of the stimulus.

Throughout learning (as well as at test) quantisation error is used as a proxy for looking time. This is in line with the notion that longer looking times during familiarisation reflect the larger degree of adjustment to the mental category representation that is necessary for items perceived to be more novel (consistent with Sokolov, 1963, and see Mareschal et al., 2003, for evidence that infant looking times reflect this on a trial-by-trial basis). In order to reflect this larger degree of adjustment, the learning rate in our model is modified accordingly. At test, quantisation error is still taken as a proxy for looking time (the model is not trained further at that stage). Even though of our approach using the dynamic learning rate is a simplification, we believe it is nevertheless a plausible way of incorporating the fact that infants' looking time is not uniform. While we acknowledge that there may be even more nuanced ways of doing this, we believe that this is a valid and persuasive approach precisely because of its simplicity. Incorporating Kidd et al.'s findings that intermediate novelty is preferred would involve a learning rate that is maximal for stimuli of intermediate novelty. Whether such an approach could capture the infant data we present here is an interesting empirical question, but we believe that that goes beyond the

scope of the paper and should instead be done as part of future work.

- I'm not sure what order the work was done in (i.e., modelling first then empirical studies or vice versa). One possibility to make the logic of the paper clearer would be to present the empirical study first, then the model. This would allow the authors to argue that the critical, novel mechanism in the model that allows it to capture the empirical data is the adaptive learning rate; this would then have implications for theories of infant learning mechanisms. However I would not insist on this if the authors can clarify in some other way.*

Reply II.2:

The order of work was (1) simulations of Mather & Plunkett's results, (2) simulation with new sequences incorporating both euclidean distance and final stimuli, (3) infant experiments to validate effects predicted by the model. We do think it is important to present the work in that order, because we believe that an important role of computational models is to create novel, testable hypotheses and this is exactly what has happened here.

From the reviewer's comments it became clear to us that this chronology isn't clear from our manuscript. We have therefore changed the structure and are presenting first a section on our rationale, describing how the hypothesis of recency effects was derived. This is followed by a section on simulations, with an introduction of the model, followed by preliminary simulations on Mather & Plunkett's data as Simulation 1, and by main simulations on recency effects as Simulation 2. Infant studies are then presented in a separate section ("Infant Experiment"). Some parts regarding sequence design and stimuli have been moved to correspond to this structure. We hope that this has clarified the manuscript.

- On page 9 the authors state: "Auto-encoders are trained by presenting all input patterns many times, and a core assumption is that the aim is to place equal weight on each stimulus, disregarding the sequence in which stimuli were presented (in other words, this approach is by definition unsuitable to model the particular memory effects we want to address)." In work from my own group (Twomey & Westermann, 2017; Dev. Sci.), we showed that an autoencoder could capture the results in M&P; so, autoencoders are also subject to sequence order effects. Whether they could capture the recency effects is a very interesting avenue for further research. The fact that two models have captured the same data doesn't impact on the novelty of this paper, which I think makes a strong and interesting contribution. More broadly, in our paper we make some complementary arguments concerning the dynamic nature of categorisation that could strengthen the discussion of the current MS. In particular, we argue that plasticity of the network is key during learning; it seems to me this relates to the dynamic learning rate employed here. I'm usually very reluctant to ask authors to cite my own work but in this case I think it's appropriate.*

Reply II.3: We thank the reviewer for this comment – the simulations in Twomey & Westermann (2017) are naturally very relevant, and we now include the findings in the revised paper (p.8 paragraph 1, p. 11 paragraphs 2 and 3).

While our aim was to justify our use of self-organizing maps we appreciate the achievements using autoencoders, which we think make an important contribution to the field. In fact, while we believe that model choice is ultimately down to subjective preference, at least in the present context, it appears that complementary approaches to simulating familiarisation studies in the end lead to converging results, as is nicely illustrated by the work presented in Twomey & Westermann's results compared to our own presented here, and overall that seems like a confirmation of the approach.

We have revised our comparison to auto-encoder models to simply highlighting the differences between the approaches on p. 11.

4. *Although in the first set of simulations (Mather & Plunkett replication) the model shows greater error in the high ED condition, as per the stronger categorisation in that condition shown by infants in the original study, the preference scores in both conditions are less than 50%. Were they below chance? If so, how should this be interpreted? The results from the second set of simulations are quite different and above chance (although these t-tests aren't reported either). Presumably this is due to the final stimulus manipulation; it would be great to see some discussion of this.*

Reply II.4: We thank the reviewer for pointing this out. The numbers reported in the preliminary simulation (now Simulation 1) were in fact preference scores with regard to the average item 3333 (i.e. looking at 3333 divided by looking at 3333 + peripheral item, which is consistent with Mather & Plunkett's reported metrics). By contrast, numbers reported in the main simulation are "novelty preference scores" for peripheral items 1111/5555 (i.e. looking at 1111 or 5555 divided by looking at 1111+3333 or 5555+3333). We decided at a later stage that in order to be able to talk about novelty preference rather than familiarity preference, this way of reporting made more sense for our purposes, but failed to adjust the numbers reported in the preliminary simulation accordingly. The revised version of the manuscript contains the adjusted numbers (p. 16, bottom paragraph), which are of course the old numbers subtracted from 1, as looking at 3333 and looking at the peripheral item sum to 1.

5. *There are places where the logic of the paper feels circular. For example, on page 6, the authors set out the hypothesis that categorisation results are due to memory and priming based on inspection of the model and comparison with infant data, then at the bottom of page 7 state that they test this hypothesis in the model. Moving this discussion on page 7 to the GD could help. Then, on page 15, the authors state "To test the hypothesis that preferential looking at test is driven primarily by recency effects we set up a new set of simulations. The aim was (a) to understand the mathematical underpinnings of the recency effects" – but at this stage, there are no recency effects to simulate. Overall it wasn't clear where they recency effects hypothesis comes from. Similar to my first question, it's possible that this could be dealt with by restructuring the paper.*

Reply II.5: We believe this ties in with misunderstandings raised and addressed in RII.2 above. We believe this is now fixed after restructuring the manuscript. Parts of two paragraphs from p. 7 were further moved to the General Discussion, p. 30 (second sentence of section) and 32 (last sentence).

The section “Rationale: Hypothesizing recency effects” (p. 8) now provides an informal introduction to our investigation by stating the order of events (preliminary simulations, new design: simulations, new design: infant experiment). On page 16-17 we then provide a more in-depth analysis of the preliminary model that points to recency effects, and makes the transition to the main simulations clearer.

6. *The MS switches between UK and US spelling.*

Reply II.6: We have changed any US spellings we could identify to UK spelling instead.

7. *P7: paragraphs beginning “We propose” and “Our model” could be combined*

Reply II.7: In the revised manuscript these paragraphs are no longer part of page 7. Parts of these were incorporated in the General Discussion.

8. *Unnecessary exclamation mark in “(reducing the error!)”*

Reply II.8: We have removed the exclamation mark.

9. *P16: in the footnote the authors state that no noise is introduced into the system at any point. However, the networks are randomly initialised, which does introduce noise – unless I’m misunderstanding, in which case apologies!*

Reply II.9: The reviewer is right: no noise is introduced in the stimuli encoding, whereas the random weights initialization does change the initial weights distribution and the unit a given weight vector is associated to. We have made this more precise in the footnote, thank you (p. 18).

10. *Figure 5: there are no outliers in the model data so reference to this in the figure caption can be removed*

Reply II.10: We have removed the sentence about outliers from the caption.

11. *P24: There is no evidence for a preference for the peripheral stimulus on test trial 1 in the Dissimilar/Near condition ($p = .094$). This section should be rephrased to reflect this.*

Reply II.11: We have removed individual condition t-test results from the brackets immediately following the ANOVA main effect, in order to reflect the fact that the ANOVA indicates the data should not be split up. This should avoid any confusion.

However, we believe that for completeness and transparency we need to state

results for the individual conditions as they were designed, and have included these in a footnote instead (p. 26).

12. A) *P25: infants who saw the novel vs average test trial didn't show a novelty preference ($ps = .075, .092$), so references to a trend should be removed.*

Reply II.12.A): We believe it is important not to entirely dismiss results like this, and believe that readers are able to discriminate between results that are referred to as “significant” and results that are referred to as “trends”. We have conducted Bayesian single-sample t-tests to examine whether the data allow us to accept the null hypothesis, but with Bayes factors of 0.728, and 0.608, respectively, this is not the case. Infant data are inherently noisy and we do not want to withhold from the reader that these are not the kinds of results that allow us to accept the null hypothesis. We have, however, rephrased this for clarity (p. 27, paragraph 1).

B) This is interesting in itself – later the authors argue that a null result at test in this paradigm could reflect a particular category structure, rather than a lack of learning. Is this what's happening here?

Reply II.12.B): It's difficult to interpret these results, after all we are talking about Test trial 3 and 4 and these occur a long time after familiarisation, so each individual participant's mental category representation has by that stage been affected by their own preference for certain test stimuli over others (in addition to any differences that may already arise by the end of familiarisation due differences in looking time). However, we do believe that infants in this study form a mental category representation, and yes, we do believe that behaviour on these trials reflects the location of the category centroid.

To some extent it is possible to work through the logic of how looking behaviour on different test trials affect this category representation. However, on the whole this is speculative at best, and we therefore choose to refrain from overinterpreting the results in the paper.

13. *P27: re. the possibility of a primacy effect in the infant data, this could be investigated with the model, although this may be outside the scope of the current MS*

Reply II.13: This is correct, we could model a primacy effect (for instance, with a very steeply decreasing learning rate). However, since we argue that the analysis of experimental data points to a recency effect so including this is both outside the scope of this paper and would probably be confusing for the reader.

III. Replies to Reviewer 2:

1. *The conceptual novelty for which the authors argue (e.g., p 4 bottom) that “category activation does not merely involve invoking a preexisting abstract representation, but is inherently linked to memory” is in fact not very novel at all. Many papers have been written about the role of memory systems in category learning (see e.g., the work by Gregory Ashby), and this assumption is central to virtually all connectionist models (and there have been many on infant and adult categorization over the past 20 years, see e.g., the work by Mareschal and colleagues for infants and models by Rogers & McClelland for adults). There is also modeling work by Westermann & Mareschal (2012) that explicitly links infant category learning to the complementary memory system approach, with exemplar learning in one component and more global, long-term memory learning in another component. This work seems relevant here but is not mentioned.*

Reply III.1. We acknowledge that the notion of linking memory and category learning is not novel in a general context, and are taking on board the reviewer’s suggestions to include more references. Westermann & Mareschal’s (2012) model is now reviewed on p. 6, and we are now also referring to some of Ashby et al.’s work (p. 5, paragraph 1). However, we believe that our contribution is important and novel in the sense that in the context of infant familiarisation the role of memory in on-line categorisation processes has not been highlighted in the literature.

2. *The authors compare their SOM-based approach to the more widely used auto-encoder models of modeling infant category learning (p 9), but in doing so they misrepresent auto-encoder models. First, they state that auto-encoders are trained by presenting all items in a familiarization set many times with the core assumption that the aim is to place equal weight on each stimulus, and that these models therefore disregard the sequence in which stimuli are presented (the key concern in the present paper). However, most auto-encoder models tend to exactly replicate the familiarization phase in infants, presenting each item only once (e.g., French, Mermillod, Mareschal, & Quinn, 2004; Westermann & Mareschal, 2004). In fact there was a recent auto-encoder model (Twomey & Westermann, 2017) that modeled precisely the order effects found in Mather & Plunkett (2011) like in the present paper, and which the authors claim cannot be accounted for in such models.*

Reply III.2. It is correct that many of the models pointed out here do maintain the sequential relationship, although items are typically presented many times in a row (e.g., 250 epochs in French et al., 2004). We have substantially revised the part of the manuscript that compares SOMs and autoencoders to reflect this (p. 11), and also included French et al. (2004), and Twomey & Westermann (2017) in several places. Please also see our reply to Reviewer

1, Reply II.3., above for corresponding changes.

3. *Second, the authors argue that analyzing the internal representations of auto-encoders is more complicated than for SOMs and imply that such analyses in auto-encoders have little or no explanatory power (line 51). This claim is surprising as there are many papers that have used clustering of hidden representations and distance measures between the hidden representations for specific inputs, which is a straightforward process (e.g., McClelland & Rogers, 2003) that has often been used to explain the internal mechanisms of learning. It is not clear why the distance between multidimensional hidden representations in an auto-encoder should have less explanatory power than the distance between multidimensional BMU vectors on a feature map. The comparison between SOMs and auto-encoders should therefore be more nuanced, and the results from the Twomey & Westermann paper, which modeled some of the same data as in the present paper, should be discussed and related to the present work.*

Reply III.3. We agree with the reviewer's point of view here. We did not in fact aim to suggest that hidden unit representations in auto-encoder networks have less explanatory power – just that they are less directly accessible because using them typically involves dimensionality reduction, whereas the question of whether two or more patterns share a best-matching unit is immediately clear and an inherent part of the SOM. We have clarified this (p.11).

4. *The SOM algorithm used here only modifies the BMU and not a neighborhood of the BMU. This procedure will lead to the SOM to become an exemplar storage without a similarity structure across the map as is common in the typical algorithm when the neighborhood radius is gradually decreased across training. This doesn't therefore seem to be a very good category learning model since it is not able to represent internal category structure (i.e., similar category members will not be represented by neighboring map units).*

Reply III.4. Correct, this neighbourhood function is indeed a very strong simplification and it leads to a map which is not an exemplar storage (see below) but which lacks topological structure. The fact that the predictions of the model are confirmed by experiments with infants shows that the model, even very abstract and simplified, says something about the categorisation process. Considering more standard neighbourhood functions will be the objective of future work.

5. *Moreover, and importantly, this lack of neighborhood updating also make me suspect that, at least in the similar-transition sequence, but perhaps also in the dissimilar-transition sequence, there will be only one unit on the map that is the BMU for all eight familiarization stimuli. I believe this to be so because one unit will be the BMU for the first item and will have its weights adapted (rapidly, because of the variable learning rate), and all other units remain randomly initialized. It's therefore very likely that the same unit will be the*

*BMU also for the second item, and so on (at least if the stimulus feature descriptions differ sufficiently from the random weight initializations). This single BMU will, at the end of familiarization, then be strongly adapted to the final familiarization item, leading to recency effects. In fact, the authors' explanation of the recency effects (p 17) completely relies on a BMU being shared between most or all exemplars: if a different BMU responded to each exemplar, then the order in which they were presented would not matter (as, in the absence of a neighborhood function, only the BMU is updated without any effect on other units). Although the authors suggest (p 18 top) that for the peripheral test stimulus a different BMU to the last training item will be selected, this is not necessary: even if there is a single BMU shared by all training and test items, the quantization error for a peripheral item would still be higher after 'near' final training item. I expect that the model would produce exactly the same results if it only had a single unit. Importantly, these are not merely technical details: the implication of this explanation is that the model cannot ***discriminate*** between the familiarization items (since they all have the same BMU), whereas the process of categorization is based on the idea that objects can be discriminated but are nevertheless treated as equivalent, and infants have often been shown that they can discriminate between items for which they form a category. – To clarify this, at the very least the authors need to report across how many BMUs the familiarization items were distributed in both training conditions and what this means for categorization.*

Reply III.5. We have analysed the emerging internal structures, and in fact the model does not always form a single-unit representation over the familiarisation stimuli but can regroup the stimuli to different unit representations, selecting different BMUs during the presentation of the stimuli. The process goes as follows: initially, a BMU is selected for the first stimulus, and its vector representation is updated accordingly. For subsequently presented stimuli, this best-matching unit is either updated to match the incoming stimulus or, if the incoming stimulus is distant from the BMU's associated vector, another BMU is chosen. The stimuli are therefore at least partly discriminable: those sufficiently similar, that have the same BMU, might not be discriminable but those sufficiently different, that have a different BMU, definitely are discriminable. Due to the high learning rate, each update can override previous representations, resulting in a recency effect.

6. *A problem is that infants familiarized to the training stimuli only in one of the four conditions (sim/near). And in fact this is the only group showing a looking preference at test (there is a weak trend – $p = .094$ – for the Diss/Near condition).*

Reply III.6. We understand why the reviewer is pointing this out, but we don't think this is a problem. What the ANOVA means is that the data in the Sim/Near and the Diss/Near conditions are in fact from the same underlying distribution. Treating the Sim/Near condition and the Diss/Near condition as

separate is therefore problematic.

7. *Interestingly the second test trial is the opposite of the first one, with the only effect now a preference for the peripheral item in the Diss/Far condition. The authors rightly note that this second test trial was not modeled (although it would be interesting and straightforward to do so), but it would be good to at least speculate how this result fits in with the narrative of recency effects – from my understanding it directly contradicts this narrative. The authors do discuss this result (p 26f) but their explanation critically assumes that infants have formed a category, but the fact that they did not familiarize to the training stimuli makes this assumption questionable.*

Reply III.7. A) Modelling Test trial 2: It is in fact not straight forward to model the second test trial, because infants are exposed to two items simultaneously during the first trial. Even if each individual child's sequence of looking at left and right items could somehow be incorporated in the training regime, comparison between two items is different from being exposed to just a single item. Any attempt of incorporating this would involve making assumptions in the model about what simultaneous exposure implies for infants. There could be interesting insights gained from a modelling process in this direction, but we feel that this is a new modelling project that is rather beyond the scope of the present paper as it would focus more on the effects of presenting items side by side, rather than recency effects.

Reply III.7. B) the fact that they did not familiarize to the training stimuli makes this assumption questionable: We don't follow the reviewer's logic. Firstly, in familiarisation paradigms like this a lack of decrease in looking during familiarisation (as opposed to habituation) is generally not taken as evidence for a lack of learning (see our reply to the Associate Editor, R1.2, above, for a discussion). In fact, in Younger (1985)'s original study, "only female infants in the Broad condition looked less over trials". Secondly, however, we believe there is at least some evidence for learning in three of our conditions, as infants show systematic preferences on test (both Near conditions on test trial 1, and Dissim/Far in test trial 2). The only alternative explanation would be that infants had a prior preference for the peripheral stimuli over the overall average, and displayed that preference after not having been affected by seeing eight familiarisation items, which seems unlikely. What about the fourth condition, Sim/Far? Interestingly there is actually a decrease across familiarisation here, but on test we see no evidence for a systematic preference.

This is not inconsistent with infants not learning in this condition. However, as we discuss in the context of the model results, it seems quite unlikely to us that infants don't learn at all in a sequence containing the same items, just in a different order – when a plausible explanation that is displayed by a computational model is that learning does take place, but with a resulting prototype that is located between the two test stimuli, leading to a null preference. To us that latter result seems much more compelling than to

assume that infants in the Sim/Far condition really didn't learn.

8. *Overall, the results from infants seem to lend only weak support for recency effects, which occurred in only one of the four conditions (and a weak trend in a second), and only on the first test trial. If anything, the data only hint at presentation order having an effect on looking at test (which was already shown by Mather & Plunkett, 2011) with the role of the final item having only a weak, if any, additional effect.*

Reply III.8. We don't follow this argumentation. We found a main effect of final stimulus type, but no other effects, and specifically not an effect of Euclidean Distance, on test trial 1. We have checked our ANOVA results and found them to be accurate. Furthermore, the pattern of results is predicted by the model. We believe we should keep our analysis and interpretation as it is.

9. *Together, these results are much less strong than the narrative makes us believe (that "perceived novelty depends mainly on similarity to the immediately preceding item" (abstract), "that the recency effect determines looking" (p 26) and that "recency effects are dominant in infant categorization" (p 27) – if anything, order effects interact with a more global learning of category structure; and that "infants exhibited a clear preference" (p 26) – they only did so in one of the "far" conditions and in no other, and only on the first test trial). The data should be discussed much more carefully and the narrative toned down.*

Reply III.9. We have revised the above phrases in the abstract and on p. 27, 28 to reflect a more careful interpretation.

10. *The conclusions that familiarization sequences should be taken seriously is an important one, as shown in Mather & Plunkett (2011), but it is typically addressed by familiarizing different infants with different sequences.*

Reply III.10. The reviewer is right that this is typically addressed by randomising the sequences. However, we also think it is important to present a systematic investigation of these effects and make this point in the literature. At no point are we suggesting that experiments not explicitly incorporating a factor of final stimulus item are somehow at fault. On the contrary, what our findings point out is that for some experiments, the impact of these final familiarisation items might obscure otherwise systematic behaviours.

11. *The issue of p values: the looking preference result for the Diss/Near condition comes out with $p=.094$ (p 25) and is interpreted as a "clear preference" (p 26); the result for looking time collapsed across conditions (p 27) comes out with $p = .097$ and is described as not differing from chance. It would be better to adopt a consistent approach to describing results as significant or not (ideally, the conventional 0.05). Of course frequentist statistics have their flaws, and perhaps the results should be supplemented by Bayesian analyses that also allow for interpretation of null results.*

Reply III.11. We believe this is actually a misunderstanding of our argumentation now on p. 28 (paragraph 3) – we agree that the wording was not entirely clear. The two-sample t-test result for preferences collapsed across conditions, which we describe as “not differing from chance”, obtained a p-values of $p=.821$. The single-sample t-test against chance in the similar transition sequences condition obtains a trend ($p = .097$). We have re-worded and expanded that passage in order to make this more straight forward. We believe we have systematically treated values between .05 and .1 as “trends” in the manuscript, and only p-values smaller than .05 are treated as “significant”.

However, we agree that a Bayesian approach might help to interpret some of these ambiguous results.

Results using Bayesian one-sample t-tests

We conducted Bayesian t-tests on the data that Reviewer 2 seems to consider unclear, in particular in order to investigate whether there is enough evidence in any of the cases to accept the null hypothesis. As sections a.-b. below show, while the collapsed data for both “near” conditions offer enough evidence to reject the null hypothesis (i.e. infants show a preference for the peripheral item), the individual condition result (Diss/Near) condition offers neither enough evidence to reject or accept the null hypothesis. Since we have responded to Reviewer 1’s request to not split the data in this section, our current version of the manuscript still lists the single-sample t-test result in Footnote 4, but it is not mentioned in the main text to reflect the fact that the “near” conditions should not be split up.

Section c. below shows a Bayesian one-sample t-test for the collapsed data on Similar conditions, which we report in the section “Discussion of Experimental Findings” as a trend. We think this is appropriate, given that the Bayesian test shows that the null hypothesis cannot be rejected.

a. Proportion of looking at the peripheral test item in both Near conditions, collapsed

$BF_{10} = 5.929$, which implies sufficient evidence to reject the null hypothesis.

Bayesian One Sample T-Test

	BF_{10}	error %
Test1Peripheral	5.929	6.857e-7

Note. For all tests, the alternative hypothesis specifies that the population mean differs from 0.5.

b. Proportion of looking at the peripheral test item in the Diss/Near condition (reported t-test result $p = .094$)

$BF_{10} = 0.784$, which implies insufficient evidence to accept the null hypothesis and insufficient evidence to accept the alternative hypothesis.

Bayesian One Sample T-Test

	BF₁₀	error %
Test1Peripheral	0.784	6.240e-5

Note. For all tests, the alternative hypothesis specifies that the population mean differs from 0.5.

c. Proportion of looking at the peripheral test item in the collapsed Similar conditions (reported t-test result $p = .097$)

BF₁₀ = 0.586, which implies insufficient evidence to accept the null hypothesis and insufficient evidence to accept the alternative hypothesis.

Bayesian One Sample T-Test

	BF₁₀	error %
Test1Peripheral	0.586	4.618e-6

Note. For all tests, the alternative hypothesis specifies that the population mean differs from 0.5.

12. *The collapsed results reported on p 26 show no effect for both the similar and dissimilar-transition conditions, although to my understanding this would be a direct replication of Mather & Plunkett (2011). Is this non-replication of the earlier results a concern?*

Reply III.12. In fact, this is not a replication of Mather & Plunkett's (2011) results since they used different sequences. The discrepancy between the results is therefore not a concern.

To explain: Mather & Plunkett constructed all possible stimulus sequences and then chose the ones that minimised / maximised Euclidean Distance. By contrast, we had a second constraint, requiring sequences of similar euclidean distance half of which ended in "near", and half of which ended in "far" items. As a result of maintaining equal euclidean distance across Similar/Near and Similar/Far, as well as Dissimilar/Near and Dissimilar/Far sequences, respectively, our sequences are overall less "extreme" in terms of euclidean distance (our "Similar" sequences traverse slightly higher Euclidean distance than Mather & Plunkett's "low distance" sequences, and our "Dissimilar" sequences traverse slightly lower Euclidean distance than Mather & Plunkett's "high distance" sequences). This is now explained in Footnote 5.

13. *P 8 line 25: the link between novelty/familiarity/no preference and 'model quantisation error' should probably be explained here (or should be rephrased).*

Reply III.13. We have amended this section in the paper and hope it is now clearer.

Appendix B

Royal Society Open Science Manuscript ID RSOS-200328.R1
Responses to editor & reviewers

I. Replies to the Editor:

- 1. Your paper has now been seen by the two reviewers that provided initial feedback - they both find your manuscript much improved, with clearer justifications and methodology and I concur with their views. I particularly liked how by restructuring the manuscript you convey the dynamic interplay between computational and empirical work. Reviewer 2 still requires some additional clarifications which you will have to provide when re-submitting your manuscript.*

Reply I.1: We thank the editor for these comments. We hope that with the alterations detailed below and marked in colour in the manuscript, the paper is now ready for publication.

II. Replies to Reviewer 1:

- 1. This is a review of a manuscript I have reviewed once previously. I apologise to the authors and editor for the delay in getting the review back to them. The authors have addressed my comments thoughtfully and thoroughly. The rationale for and chronology of the studies are now much clearer, and the authors now provide a more nuanced discussion of autoencoder vs. SOM approaches. The conclusions the authors draw are now more conservative, and my major questions have been addressed. On this basis I have no further questions. I would however ask that the authors share their model and analysis code, as per the journal's requirements.*

Reply II.1: We thank the reviewer for their comments. The model and analysis code have already been made available at the Open Science Foundation at <https://osf.io/zt3fd>, as noted in the manuscript (Data Accessibility Statement).

III. Replies to Reviewer 2:

- 1. The paper is now much improved and the authors have done an excellent job of addressing many of the raised points in detail. Most of my remaining points concern improving the transparency of the model. I do believe that the authors need to be more explicit about the contribution made by the model to explaining the recency effect. Relatedly, the comparison between the present model and more typical SOMs lacks detail. Specifically:
The comparison between better known standard SOMs and the current model is lacking in detail. Clearly, the lack of a neighborhood radius in the model is a major departure from standard SOMs because without it, no topologies are formed, and these are a core property of standard SOMs. In the discussion of*

the differences between the current model and a typical SOM (p10), this should be mentioned. Again on p 15, the authors state that the main difference between theirs and standard SOMs is the adaptive learning rate, and the lack of a neighborhood radius is not explicitly mentioned but just described by the Kronecker function (p. 15) – this important detail will escape most readers not familiar with SOMs. It is therefore important that the lack of a neighborhood update scheme is spelled out clearly as a difference to typical SOMs, as it greatly affects the functioning of the model.

Reply III.1: Thank you for pointing this out. This is correct, and we have added a more detailed comparison with standard SOMs on p. 10 (paragraph 1) and 15.

2. *The explanation of how the model learns categories on p 10 relies on a non-zero neighborhood function ('.. because that unit has been moved repeatedly towards familiarization exemplars' and '...because no unit has been moved in that direction'). While this explanation is correct for typical SOMs with neighborhood updating, the explanation in the zero-neighborhood model (which does not develop topological representations) is rather different, namely, that a similar item shares the same BMU with training exemplars, and a dissimilar item might not. It should be made clearer in the paper that typical SOMs and the current model learn categories in these different ways.*

Reply III.2: We don't agree that there is a contradiction here – our case is merely a special case of general SOM learning. General SOMs certainly don't prevent a scenario in which the test item activates the same BMU as a familiarisation item. The second part of your point is accurate, our model works on the basis that the test item activates a unit that was also the BMU for at least one familiarisation exemplar. Since training in our model almost always involves several different BMUs (see reply III.3 below), the selection of the BMU at test still relies on the best match out of multiple possible units. We have rephrased our description on p. 10 to make this clearer ("The expectation is that a within-category stimulus will be relatively close to an existing unit, because one or more units were moved into its direction during training as BMUs of familiarisation exemplars. In other words, a within-category stimulus might share a BMU with one or more familiarisation stimuli. The result is that the within-category stimulus will not generate a large quantisation error."). We believe this passage now captures precisely what happens during learning (whereas during standard SOM learning, a unit might be selected even though it never acted as a BMU).

3. a) *In the response letter the authors state that not all of the models had only one activated BMU – does this mean that most did? It should be reported over how many units the BMUs are distributed on average.*

Reply III.3a): In the simulations we considered, 64% of the cases used 2 different BMUs, 29% used 3 BMUs, 4% used 4 BMUs and only in 3% of the cases there was only one BMU. We have included these results on p. 19/20,

and also added statistical tests to assess differences across the conditions, as well as a passage in the Discussion section to discuss this (p. 21, paragraph 2).

b) As I wrote in my review of the previous version of the paper, the recency effects (and order effects in general) in the model in my view crucially depend on the sharing of the BMU by most or all training and test exemplars, since there is no neighborhood updating. In my view this means that recency effects arise because the model does not discriminate between different category exemplars – if two stimuli share a BMU then the model does not discriminate between them. The authors might disagree, but this should be discussed. To my reading this prediction – that recency effect arise because the model does not discriminate between specific stimuli - is the main contribution of the model as it is a clear prediction that can be tested.

Reply III.3b): We thank the reviewer for this interesting comment. Firstly, as most of the simulations involve several BMUs, it is not the case that the model does not discriminate between the training exemplars, even if we adopt the reviewer’s definition of “discrimination”. Secondly, in our view it is not accurate to say that the model does not discriminate between two patterns if they activate the same BMU. If the model treated two patterns the same, the resulting quantisation error (the model equivalent of looking time) would be the same. Here, the quantisation error between different patterns can be different, even if two familiarisation stimuli (or indeed a familiarisation stimulus and a test stimulus) activate the same BMU. We would argue that the number of BMUs formed is instead an indicator of the internal structure of the category being formed, e.g. whether there is recognition of any clusters within the category.

4. *On p 19f the authors explain intuitively how recency effects arise in the model. They state that for a far test stimulus after a near final training item a different BMU will be selected – was this the case in the model? It would improve the transparency of the model if this was analyzed.*

Reply III.4: This varies across sequences. We have clarified this on p. 20. To explain learning in the model further: The fact that one item presented at test will activate a unit that has also been the BMU for one or more familiarisation item is not a built-in effect, but the consequence of the fact that the items used here are not random: one of them is the overall average and therefore has a high likelihood of sharing a BMU with one or more familiarisation exemplars (let’s call this the Near-BMU); this is only overridden if one exemplar has been presented that is highly similar to the remaining test item and its BMU (the Far-BMU) was moved further towards that remaining test item than the Near-BMU was moved towards the overall average. That this is the case if such an exemplar (the “far” item) was presented last is a consequence of the adaptive learning rate. It does not matter at this stage whether the Far-BMU and the Near-BMU are in fact the same unit, or whether they are two different units – what matters is that at testing stage, if a “far” item has just been

presented the category representation is shifted towards that item (but if a near item has just been presented the category representation is very close to the overall average).

5. *In Simulation 1 the order effects from Mather & Plunkett (2011) are replicated, but the authors also state that the final familiarization stimulus had an effect. Were these effects independent, or were they driven by recency effects? That is, were the dissimilar-transition sequences more likely to end in a 'near' item than the dissimilar-transition sequences? This would be important to know to understand whether the model formed more global sequence representations or if it was entirely driven by local transitions between the final training item and the test item. Perhaps the authors can add some detail about this at the end of that section.*

Reply III.5: We thank the reviewer for highlighting this point and agree that that was not clear enough in the manuscript: In Mather & Plunkett's (2011) stimuli all similar-transition sequences ended in "far" items and all dissimilar-transition sequences ended in "near" items, as a consequence of choosing sequences with the highest/lowest possible transition similarity. This was one of the main motivations for the new simulations; we have now clarified that on page 17.

6. *It would be useful to report (and interpret) the Bayes factors that were calculated in the response letter also in the paper.*

Reply III.6: We have included the Bayes factors in the paper (p. 27 and 30).

Please note: Beyond the changes in response to the reviewers' comments listed above, we have made stylistical changes to some of the to improve clarity. These are marked in the text in colour, but do not affect the content or findings of the paper.